# Intrinsic attraction driving the high temperature performance of additively manufactured aluminum alloys

Yueting Wang [1], Chengzhe Yu[1], Kefu Gan [2] ✉, Tiechui Yuan[1] & Ruidi Li [1,3] ✉

Developing additive manufacturing (AM) aluminum alloys with high temperature strength remains a formidable scientific challenge, primarily due to the strengthening precipitates coarsening above 200°C. Conventional heat-resistant alloy design strategies aim to hinder the precipitate coarsening by incorporating low diffusive alloying elements. However, such approaches remain ineffective against thermally driven defect mobilization, especially for vacancy diffusion and dislocation climbing, which are dominant drivers of high temperature weakening. As a result, most AM Al alloys exhibit a rapid decline in strength within this critical temperature range. Through reverse-engineering of intrinsic atom-defect/atom attraction, we employ an intrinsic attraction (IA) strategy to trigger multi-dimensional defect confinement mechanisms. This approach achieves: divacancy clusters anchoring free vacancies; solute atmospheres capturing mobile dislocations and suppressing creep deformation; specific segregation forming nanostructures at precipitate interfaces and interiors to inhibit coarsening. The AM heat-resistant Al alloy demonstrates satisfactory high temperature performance, exhibiting yield strengths of ~305 MPa at 300°C, ~190 MPa at 400°C, coupled with creep resistance at 200-400°C ($\dot{\varepsilon} < 10^{-7}$/s) and prominent processability for large-size bladed disk. This strategy transcends the conventional empirical paradigm by engineering elemental segregation tendencies at specific sites, provides a universal design approach for the development of aluminum alloys or other high temperature structural materials.

Modern aircraft necessitate enhanced high temperature performance of lightweight alloy components. Additive manufacturing aluminum (AM-Al) alloys have been recently integrated into numerous key aerospace components, primarily due to their low density, high specific strength, and good manufacturability[1]. However, commercially available AM-Al alloys (e.g., Al-Si and Al-Mg systems), despite exhibiting good room-temperature strength, are generally limited to service temperatures below ~150 °C[2] owing to their severe strength degradation at the

required 300–400 °C range. Such strength reductions primarily stem from the inevitable coarsening of precipitates, which are the key strengthening mechanism in most Al alloys[3,4]. On the other hand, crystal defects (e.g., dislocations) become profoundly active at high temperature regimes, which can hardly be inhibited by coarsened precipitates and further lower the resistance against external stresses.

Significant efforts have been devoted to improving the heat resistance of AM Al alloys. However, most are centered on low diffusive

[1]State Key Laboratory of Powder Metallurgy, Central South University, Changsha, China. [2]School of Materials Science and Engineering, Central South University, Changsha, China. [3]National Key Laboratory of Science and Technology on High-Strength Structural Materials, Central South University, Changsha, China. ✉e-mail: gankefu@csu.edu.cn; liruidi@csu.edu.cn

alloying design strategy, specifically the manipulation of in-situ formed micro or nano scale precipitates. For instance, interfacial segregation of slowly diffusing solutes, like Zr[5] or Mn[6], was applied in Al-Cu systems. Partially delayed coarsening of the nanoprecipitates failed to develop higher strength, with the alloys exhibiting less than 100 MPa at 400 °C. Similarly, interwoven networks of eutectic precipitates consisting of low diffusive elements (e.g., $Al_3Ni$[7,8], $Al_{11}Ce_3$[9,10], $Al_6Fe$[11,12]) were meticulously engineered along grain boundaries (GBs) and melt pool boundaries, with precipitate sizes ranging from 100 to 500 nm[13,14]. However, the alloy still inevitably suffered noticeable strength degradation resulting from rapid phase coarsening above 350 °C. Apart from this, some endeavors have completely discarded the obsession of precipitate-coarsening control. Instead, they directly replace the self-precipitated phases by adscititious high-melting-point ceramic particles[15,16]. However, a great mismatch of modulus and thermal expansion exists between the Al matrix and such ceramic particles, unfortunately causing a significant dilemma of substitutionally compromised processability and toughness.

To solve this problem, we need to reconsider the intrinsic origin of high temperature failure in precipitate-strengthening AM-Al alloys. Specifically, the fundamental mechanisms of such a failure can be ascribed to thermally driven defect mobilization, and this manifests into two key consequences: on the one hand, mass transfers between coarsening precipitates and the surrounding matrix critically depend on anabatic vacancy-mediated diffusion of precipitate forming atoms. On the other hand, accelerated dislocation slip and climb promote the creep of Al alloys at elevated temperatures. The prior studies predominantly concentrate on retarding precipitate coarsening or introducing thermally stable second phases, yet the intrinsic role of defect dynamics in governing high temperature failure remains insufficiently addressed.

In this work, to counteract the inherent defect motion, we proposed a strategy based on pre-designed element segregation tendencies at specific sites. This approach aims to amplify intrinsic attraction among specific atoms and defects, thereby capturing them from multiple dimensions, including vacancies (0D defects), dislocations (1D defects), and precipitate forming atoms (constituting 2-3D features). Ultimately, this strategy enhances both the strength and thermal stability of the matrix and the precipitates. As a result, the optimized heat-resistant and high strength AM Al alloys (called IA alloy here) demonstrate strong high temperature performance, exhibiting yield strengths of 305 MPa at 300 °C and 190 MPa at 400 °C, as well as a good creep resistance at 200-400 °C. Departing from the conventional paradigm of deducing strengthening mechanisms from properties, we reverse-engineered the process by pre-defining the underlying mechanisms to achieve high temperature strength. Furthermore, the core concept of amplifying intrinsic attraction offers broad applicability across diverse metal material systems.

## Results and discussion
### Intrinsic Attraction Strategy

Against thermally induced defect mobilization, we seek to anchor defects (e.g., vacancies, dislocations) by exploiting strong intrinsic solute-defect attractions, preserving enough immobile defects at elevated temperatures, as shown in Fig. 1a. Crucially, the inherent rapid solidification characteristics of AM, particularly the high cooling rates ($10^5$-$10^7$ K s$^{-1}$) in powder bed fusion-laser beam (PBF-LB) processing, enable the formation of supersaturated solid solutions and substantially expand the window of element screening. This allows low-solubility but promising solute elements like Fe, Cr, Ni, and Mn (Supplementary Fig. 1) to be candidates, overcoming the limitations of metallurgical solubility in wrought counterparts. As forementioned, the intrinsic attraction (IA) strategy for solute element screening is manipulated in 3 different dimensions:

Vacancy, a zero-dimensional defect, is generally not considered to provide strength. Instead, it even offers a lower energy barrier channel for atomic diffusion. Although the lifetime of a single vacancy is indeed too short to play a major role, free migration of multiple vacancies promotes non-equilibrium solute atomic segregation and precipitate coarsening[17,18]. Moreover, dislocation climb, driven by vacancy diffusion through the dislocation core or from the surrounding lattice[19], enhances the creep deformation at elevated temperatures. Consequently, conventional wisdom holds that minimizing vacancy density or suppressing their long range motion is key to preserving high temperature strength. Notably, AM alloys inherently possess supersaturated vacancy concentrations ($\sim 10^{-4}$ at%, data from this work), already comparable to those at the melting point of most metals[20]. This concentration is at least one order higher than cast alloys due to the complex Marangoni effect and rapid solidification, and such excess vacancies are essentially unavoidable. Given this inherently high vacancy concentrations, simply reducing their concentration is ineffective. Therefore, our strategy shifts from attempting to eliminate vacancies to immobilizing them by anchoring these defects and suppressing their long range migration. For this purpose, the intrinsic attraction is expected to originate from specific elements that can strongly combine nearby vacancies to form long lifetime solute-vacancy complexes[18]. Throughout this work, binding energies are reported as positive quantities using $-E_{binding} = E(Al_{N-2}X_1V_1) + E(Al_N) - E(Al_{N-1}X_1) - E(Al_{N-1}V_1)$[21,22]. It should be noted that the minus sign is used to make the solute vacancy binding energy calculated in this article positive: positive binding energy indicates favorable binding. Among all candidates (Fig. 1b), the Cr-vacancy complex shows the strongest binding energy of 0.34 eV. Cr is therefore selected to anchor high concentration solute-vacancy clusters for effective matrix stabilization. All binding energies reported in this work follow this positive convention.

(ii) Dislocations, which are one-dimensional defects in Al alloys, act as the main source of high temperature softening, resulting from the activation of climb and glide behaviors at elevated temperatures. Since dispersed precipitates effectively hinder planar slip but are insufficient to inhibit dislocation climb[23], alternative strategies are required to counteract this specific dislocation mechanism. It is noteworthy that certain solute atoms exhibit strong tendency to segregate around an edge dislocation, forming persistent solute atmospheres that can dynamically drag migrating dislocations[24]. Based on this, our calculations of the intrinsic attraction between solute atoms and edge dislocations (Fig.1c) indicate that Fe and Cr exhibit the strongest tendency to segregate at such sites, identifying them as promising alloying elements for heat-resistant Al matrices. Furthermore, the anticipated Fe-Cr-containing clusters exhibit extremely low formation energy (Supplementary Fig. 2), indicating high feasibility of their formation.

(iii) Finally, we return to the central matter constituting 2-3D features, i.e., precipitates themselves. The degradation of alloy strength is primarily caused by Ostwald ripening-induced coarsening of precipitates, which occurs through elemental diffusion-driven dissolution and growth of particles[25]. This process can be suppressed when certain alloying elements (e.g., Fe/Cr) exhibit high binding affinity with the precipitate forming atoms, stably segregate at the precipitate-matrix interfaces. There, they form fine-scale clusters that act as barriers, hindering the free diffusion of key atoms into the precipitates. By effectively capturing these atoms, the clusters limit further growth and dissolution, thereby inhibiting coarsening. We performed binding-energy calculations on various dual-element pairs to identify candidates with the strongest intrinsic attraction for segregation. Here, the more positive value indicates the greater system stability. The result presented in Fig.1d leads to the selection of Sc and Zr. Strategic incorporation of Sc and Zr promotes binding with Fe/Cr, resulting in the formation of clusters at and within particles. This

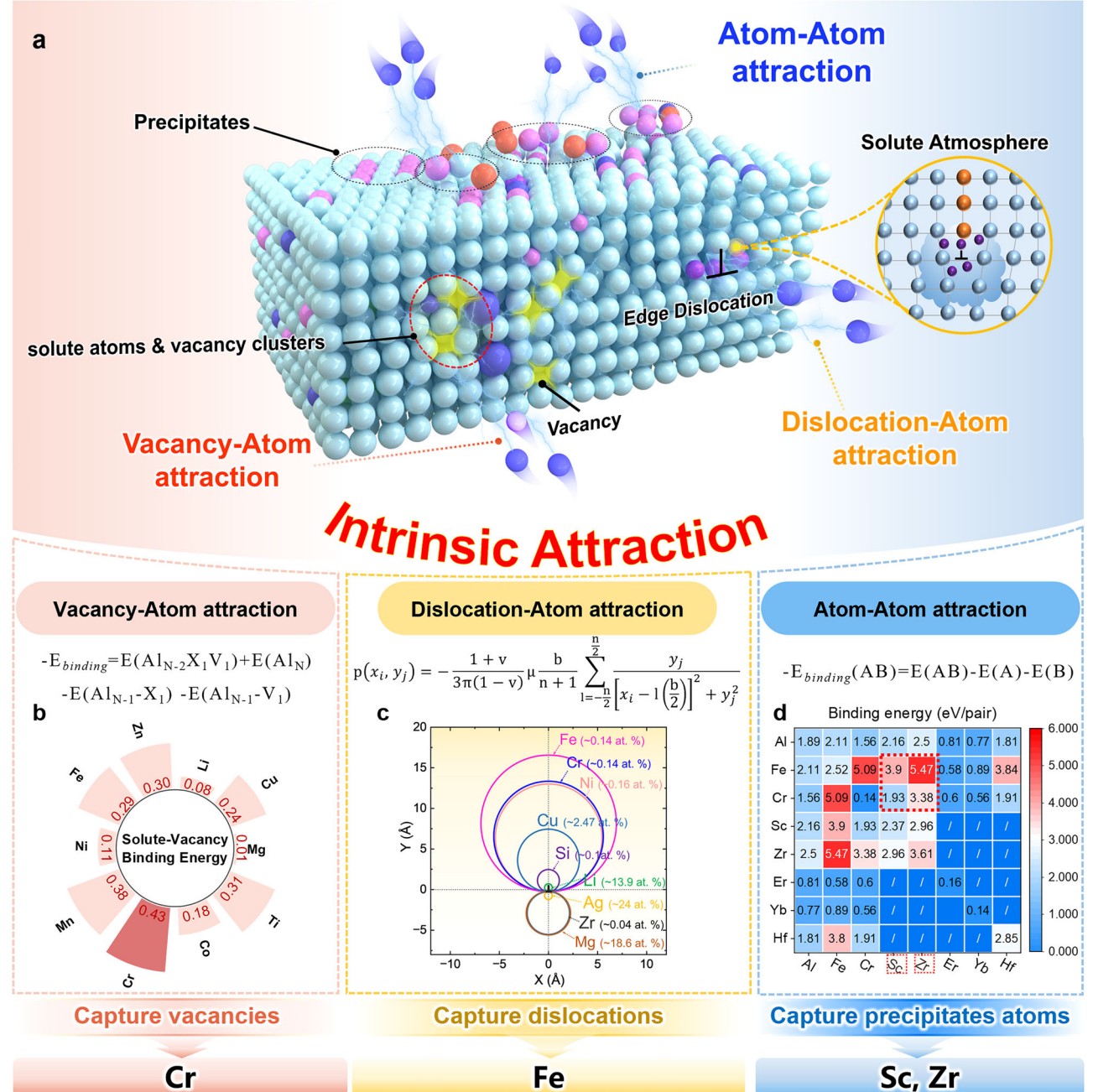

**Fig. 1 | Intrinsic Attraction design strategy. a** Diagram for the intrinsic attraction strategy against thermally driven defect mobilization. **b** Vacancies-atom attraction. Element selection based on vacancy-capturing capability. DFT-calculated binding energies identify elements with stronger attraction to vacancies. **c** Dislocations-atom attraction. Element screening through dislocation interaction analysis. Solute atmospheres around an edge dislocation in Al are visualized, where each circle represents regions with solute concentration exceeding 10 times the concentration away from the dislocation. Circle diameters correspond to the average displacement vector of relaxed atomic configurations, where larger diameters indicate stronger attraction to dislocations. The dislocation core (0,0) contains an extra half-plane in the y > 0 region. **d** Atom-atom attraction. Element selection for precipitation stabilization. Rare earth elements are prioritized using binding energy calculations with the primary precipitate forming element. Stronger atomic attraction suppresses particle coarsening.

mechanism stabilizes small-scale precipitates and thereby retaining the alloy strength at elevated temperatures.

Guided by this IA strategy, Cr, Fe, Sc, and Zr elements are selected as the key alloying elements of our heat-resistant and high strength AM Al alloy. The specific chemical composition of the Al-Fe-Cr-Sc-Zr alloy (IA alloy) was then optimized by thermodynamic assessment (Supplementary Fig. 3), and it is detailed listed in Supplementary Table 1. To further validate the effectiveness of the IA design strategy, multiple additional batches of PBF-LB Al alloys with varying compositions,

including Al-Fe-Sc-Zr, Al-Cr-Sc-Zr, Al-Ni-Sc-Zr, and Al-Fe-Cr systems, were fabricated under identical processing conditions and subjected to the same heat treatment to enable direct comparison (Supplementary Fig. 4). Details of the above-mentioned energy-based DFT calculations are described in the Methods and Supplementary Note1-3.

**Alloy performance**
To evaluate the mechanical properties of the designed IA alloy, tensile tests at ambient (~25 °C) and elevated temperatures were conducted

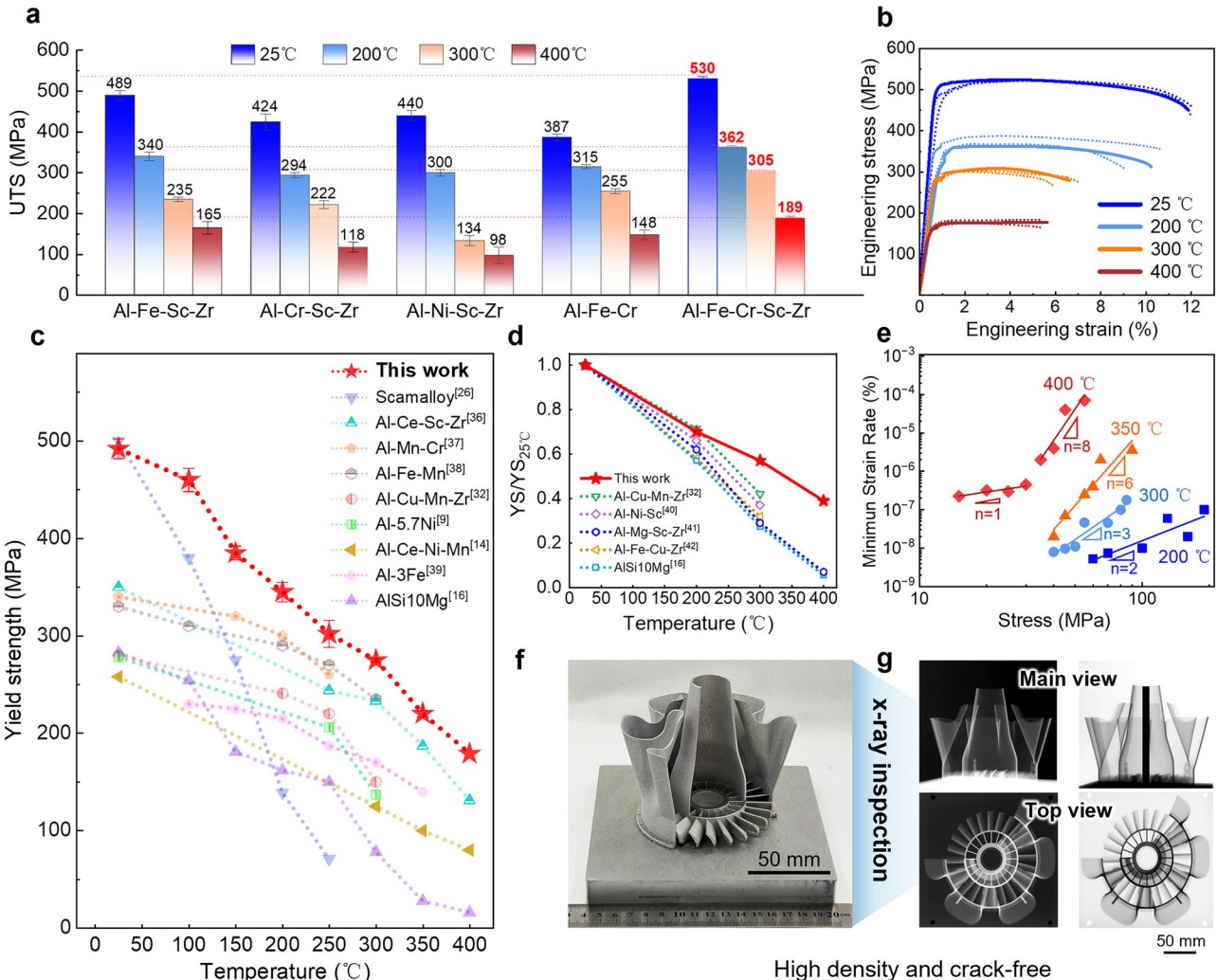

**Fig. 2 | Mechanical and formability properties of IA alloy produced by PBF-LB and subsequently heat-treated. a** The UTS of PBF-LB Al-Fe-Sc-Zr, Al-Cr-Sc-Zr, Al-Ni-Sc-Zr, Al-Fe-Cr, and IA alloys after heat treatment (HT, 325°C 4 h) were compared at different temperatures. Bars represent the mean (*n* = 3), individual data points are overlaid, and error bars denote the standard deviation. **b** Representative tensile data of IA alloy. **c** Yield strength versus temperature, alongside conventional (Scamalloy[26] and AlSi10Mg[15]) Al alloys and other reported AM heat-resistant Al alloys[8,13,31,53–56]. Each data point is the average of three measurements, and the error bars are the standard deviation of the mean. **d** YS/YS$_{25°C}$ at elevated temperatures of IA alloy compared with other L1$_2$ strengthened Al alloy after heat treatment[15,31,57–59]. **e** Double-logarithmic plots of the minimum strain rate with stress for IA alloy at 200 °C, 300 °C, 350 °C, and 400 °C. Minimum creep rates are averaged from three independent tests at each condition. **f** A 3D-printed part uses IA alloy powder on the build plate. **g** The X-ray inspection results indicated that the bladed disk is highly dense and free of cracks.

on both the IA alloy and other comparative alloys after applying the same heat treatment (325 °C for 4 h). The results demonstrate extraordinarily high strength of IA alloy, particularly at high temperature regimes (Fig. 2a, b), confirming the success of the Intrinsic Attraction design strategy. For clarity, the detailed mechanical property data for all tested alloys are comprehensively summarized in Supplementary Table 2. At room temperature, the alloy exhibits a yield strength (YS) of ~490 MPa and an ultimate tensile strength (UTS) of ~538 MPa, comparable to commercial Scamalloy® alloy[26]. Crucially, the alloy maintains high strength across temperatures from 25 °C to 400 °C, outperforming nearly all existing heat-resistant Al alloys (Fig. 2c). More notably, its UTS is ~382 MPa at 200 °C, ~325 MPa at 300 °C, and ~190 MPa at 400 °C, exceeding the UTS of AlSi10Mg at these temperatures by more than 2.5 times. Furthermore, the YS at 300 °C (295 MPa) shows only a ~38% reduction compared to its room-temperature value (480 MPa), representing the largest strength retention among current nanoprecipitate strengthening Al alloys (Fig. 2d). Pinned dislocations can be observed in both room-temperature and high temperature tensile samples (Supplementary

Fig. 5), demonstrating the effectiveness of the dislocation pinning strategy proposed by the IA approach.

The good high temperature properties of the IA alloy were further evidenced by tensile creep experiments. As shown in Fig. 2e, it achieves a low creep rate of ~10$^{-7}$ s$^{-1}$ under an applied stress of 190 MPa (~58% of YS$_{200°C}$) at 200 °C and 80 MPa (~30% of YS$_{300°C}$) at 300 °C. Based on stress exponent (*n*) values determined from log-log plots of creep rate versus applied stress, the dominant creep mechanism varies from 200 °C to 400 °C. At 200 °C, Coble creep predominates (n ≈ 2), mediated by mass transport of GB diffusion[23]. From 300 °C to 350°C, the dominant mechanism turns to dislocation glide (*n* ≈ 3-6)[27]. At 400 °C, the mechanism becomes stress-dependent: Nabarro-Herring diffusion creep (*n* ≈ 1) operates under low stresses (<30 MPa)[28], while dislocation climb (n ≈ 8) dominates at higher stress levels (>30 MPa)[29]. Here, dynamically segregated solute atmospheres are expected to form around dislocations, dragging them and thereby reducing the stress sensitivity of the creep rate (*n* < 10)[30]. Notably, the alloy exhibits a creep rate more than one order of magnitude slower compared to PBF-LB Al-Cu-Mn-Zr alloy[31] and cast Al-Ce[32] or Al-Mn[33] alloys at these

temperatures, especially when high stresses are applied (Supplementary Fig. 6). The above observation confirms the synergy of high temperature strength and creep resistance in this IA alloy. Its good creep resistance, which outperforms cast Al alloys, is mainly due to (i) stronger dislocation migration resistance from thermally stable cellular structure and (ii) high volume fraction (~30%) of submicron intermetallic strengthening phases, which are coarsening-resistant (Supplementary Fig. 7).

More importantly, along with good performance maintained at elevated temperatures, the IA alloy gains a desirable processability for large sized components, a bladed disk system with a dimension over 150 mm (Fig. 2f), which has been a critical challenge for AM heat-resistant Al alloys[34]. The results of X-ray inspection from different directions indicate that the bladed disk is highly dense and free of cracks. Although existing studies mainly focus on enhancing mechanical properties of AM Al alloys, crack-free manufacturability remains a critical yet underemphasized requirement. The IA alloy addresses this issue by enabling reliable crack-free processing (Fig. 2g) together with satisfactory overall performance.

## Multi-scale microstructure characteristics

To track the origins of such good high temperature properties, microstructural features of PBF-LB IA alloy samples were characterized across multiple scales in the heat-treated condition, matching the condition of the samples used for mechanical testing, as illustrated in Fig. 3. A bimodal grain structure is found in electron backscatter diffraction (EBSD) and transmission electron microscopy (TEM) images (Fig. 3a, b), where the solidified melt pool has columnar grains (CGs, mean size: $7.7 \pm 0.9 \, \mu m$, see Fig. 3c) inside and fine equiaxed grains (FGs, mean size: $1.1 \pm 0.26 \, \mu m$, area fraction: ~35%, see Fig. 3d) at the boundary. Inside the melt pool, the CG region is dominated by a continuous eutectic network (Fig. 3e), and the corresponding energy dispersive X-ray spectroscopy (EDS) analysis shows that such a eutectic domain consists of cellular Fe-rich phase separated by α-Al channels. The cellular Fe-rich phase is derived from the separation of the remaining liquid along the cell boundaries of initial α-Al cells, similar to the eutectic Al-Fe alloy[11,12] reported previously. There is also a minor amount of Cr, Sc, and Zr enrichment on the cell boundaries. Derived through fast Fourier transform (FFT) pattern along the [313] axis (Fig. 3f), the Fe-rich eutectic is recognized as a typical $Al_6Fe$ phase. In the FG region, a large number of Fe/Cr-rich particles form at GBs, with a size ranging from ~10 nm to ~150 nm. The EDS map in Fig. 3g also suggest a noticeable Sc enrichment inside these particles. The Fe/Cr-rich particles were identified from the XRD results (Supplementary Fig. 8) and the FFT pattern in Fig. 3h as $Al_{13}(Fe, Cr)_{2-4}$ phases, which adopt a monoclinic structure under near-equilibrium conditions. In contrast, $Al_{45}Cr_7$ (also referred to $Al_{13}Cr_2$ or $Al_7Cr$[35]) displays icosahedral symmetry, while $Al_{13}Fe_4$ (also denoted as $Al_3Fe$[11]) exhibits decagonal symmetry. Despite this difference, both are quasicrystal approximants with monoclinic space group $C2/m$ and share significant structural similarities[36]. Therefore, they are collectively classified as $Al_{13}(Fe, Cr)_{2-4}$ phases in this work.

The elemental partitioning between the precipitates and α-Al matrix was further investigated using atom probe tomography (APT). Figure 3i shows the 3D profile of the volume reconstructed with Al atom distribution (red) and iso-composition interfaces (Cr in blue; Fe in purple; Sc in orange) in the α-Al matrix. A large number of $L1_2$-$Al_3(Sc, Zr)$ particles are also formed in the α-Al matrix, evidenced by a HADDF-STEM image (Fig. 3j) showing numerous nanometer-sized bright contrast features (marked by orange arrows). They are identified as $L1_2$ (~2-5 nm) based on the FFT pattern acquiring along the $[110]_{Al}$ direction. These finely dispersed particles contribute significantly to the overall strength enhancement[37]. In particular, the $L1_2$ precipitates maintain perfect coherency with the Al matrix, as demonstrated in Supplementary Fig. 9. Figure 3k displays the 1D compositional profiles

acquired along the arrow in Fig. 3i. The profile across the Fe/Cr-rich particle and α-Al matrix shows that the particle is close to the stoichiometry of $Al_{13}(Fe, Cr)_{2.7}$ intermetallic. Interestingly, the particle shows a core-shell structure where Fe is highly rich in the particle boundary while Cr is concentrated inside the particle. Its formation can be attributed to a typical peritectic reaction observed in the Al-Cr system ($L + Al_{11}Cr_2 \rightarrow Al_{13}Cr_2$, at 725 °C)[35], which occurs earlier during solidification and then provide a nucleation site for Al-Fe compounds. Figure 3l shows an in-zoom view of the composition profile across an interface with significant Sc enrichment (~0.3 at%). The concentration of Sc and Zr (Fig. 3m) inside the $Al_{13}(Fe, Cr)_{2-4}$ particles is significantly higher than that in the Al matrix. Thus, apart from the eutectic network in the CG region, co-precipitation of thermally stable $Al_{13}(Fe, Cr)_{2-4}$ phase at equiaxed GBs and dispersed $L1_2$ phase (<5 nm) enables the high strength of the IA alloy.

## Enhanced high temperature performance through solute-vacancy clusters and stabilized nano precipitates

Vacancy trapping plays a critical role in controlling the thermal stability of precipitates in Al alloys[38,39]. To evaluate this effect, we employed the positron annihilation technique[40] to measure the vacancy concentration across several alloy samples (see Methods). The result in Fig. 4a shows that nano-grained pure Al[41], PBF-LB Al-Fe-Sc-Zr, Al-Cr-Sc-Zr, and Al-Fe-Cr alloys exhibit close positron lifetimes within 220–240 ps, indicating similar point defect types in these materials. Remarkably, the IA alloy has a longer positron lifetime (average lifetime over 247 ps) and an unprecedented level vacancy concentration compared to other alloys (see Supplementary Table. 3). Our calculations of vacancy defect fractions reveal that monovacancies and divacancies account for over 88% of these defects in the IA alloy (Fig. 4b). Further analysis of coincidence doppler broadening (CDB) spectroscopy shows that the curve of the IA alloy is quite similar to pure Cr (Supplementary Fig. 10), suggesting that positrons should predominantly annihilate in Cr-rich areas. This finding agrees well with our DFT predictions (Fig. 1b), confirming that Cr atoms preferably react with free vacancies and they generate preconceived solute-vacancy clusters. Furthermore, compared to other AM-Al alloys, the IA alloy exhibits the smallest absolute value of characteristic peak intensity, as shown in Fig. 4c. This scenario arises because more positrons annihilate at vacancies[41]. Here, the annihilation probability for high-momentum electrons is lower, causing a reduction in peak intensity in the high-momentum region of the CDB spectrum[42].

To visualize the solute clusters caused by the expected solute-vacancy reactions, we conducted an atomic-scale APT analysis across several AM-Al alloys. Representative domains were examined via a maximum separation method[43] to identify Fe/Cr-rich clusters (Fig. 4d-f). In the Al-Fe-Sc-Zr alloy (after HT, free of Cr), solute clustering was absent. We deduced that a large portion of solute atoms were collected into precipitates, rather than reacting with vacancies. The histogram of the solute 2NN pair distance (Fig. 4g) closely matches that of a random solid solution. This is further supported by the radial distribution function (PRDF) analysis in Fig. 4h, which confirms the absence of the anticipated solute clustering in this alloy. In a sharp contrast, the IA alloy has abundant clusters after HT. Specifically, Cr-rich clusters with a solute concentration twice that of the matrix, show an average diameter of ~1.1 nm and a number density as high as ~$3.03 \times 10^{25}$ m$^{-3}$. After creep deformation at 300 °C for 100 h, the cluster density in the IA alloy significantly increases to ~$8.13 \times 10^{25}$ m$^{-3}$, the intra-cluster solute concentrations raise up to 3.5 at%, the average diameter only slightly increases to ~2.1 nm. This result demonstrates the excellent thermal stability of such solute clusters. Additionally, the histogram of 2NN solute pair distance in Fig. 4i and k reveal a non-random solute segregation in both samples after HT and creep deformation. Also evidenced by the PRDF analyses centered on Fe and Cr atoms (Fig. 4j, l), the highest enrichment of Sc and Fe occurs at the first nearest

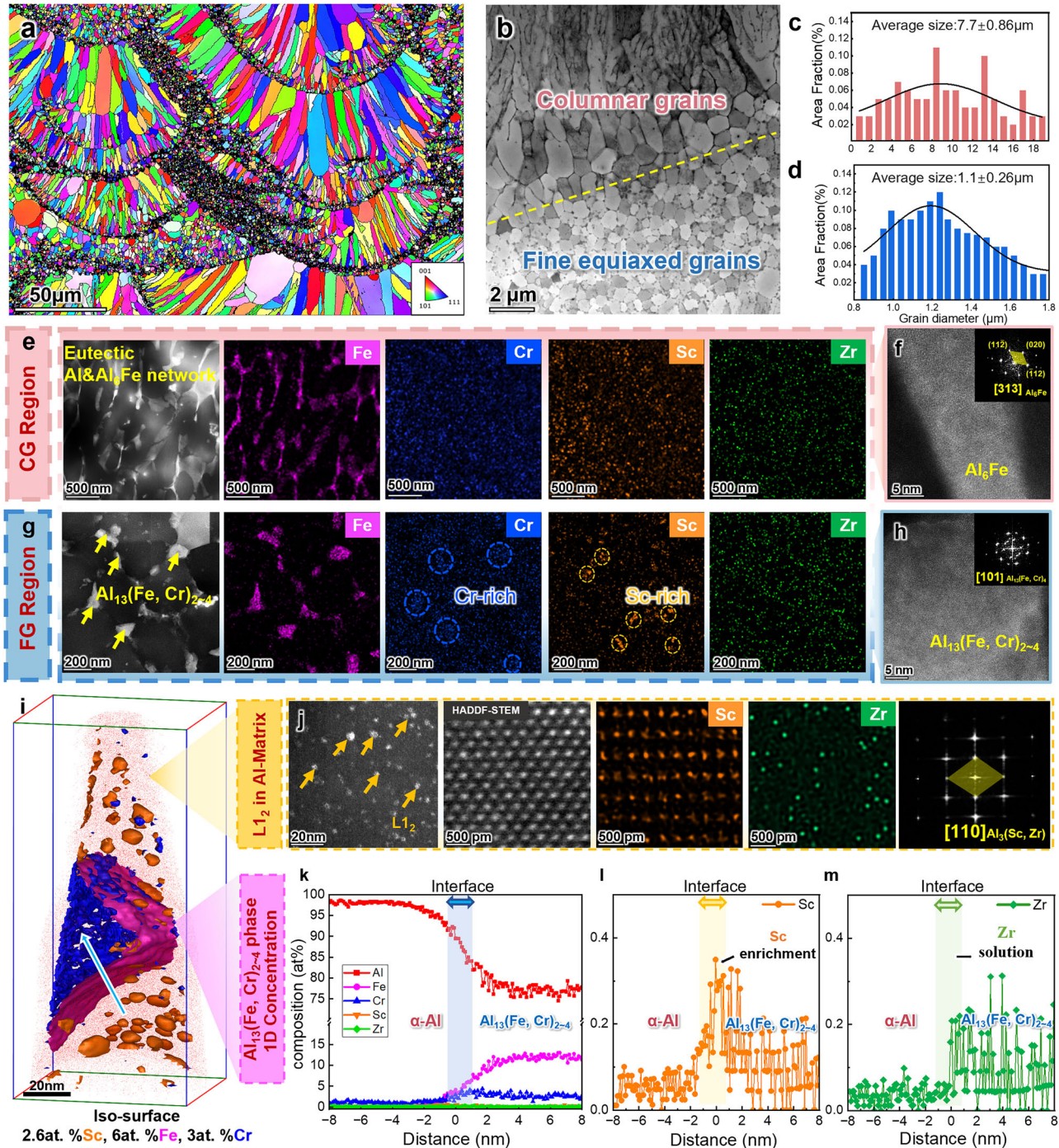

**Fig. 3 | Overview Microstructure of heat-treated PBF-LB IA alloy and APT analyses of precipitates. a** EBSD IPF micrographs showing the alternative growth of equiaxed-columnar bimodal grain structure. **b** TEM image showing equiaxed and columnar grain structure and **c**, **d** calculated grain size distribution, respectively. HAADF-STEM image and associated EDS mapping of **e** eutectic structure and **g** equiaxed grains. High Resolution Transmission Electron Microscopy (HRTEM) image showing **f** Al₆Fe network in columnar grain region and **h** Al₁₃(Fe, Cr)₂₋₄ phase in equiaxed grain region, with corresponding fast Fourier transform (FFT) image. **i** 3-D reconstruction of an APT tip (aged at 325°C 4 h) with 6 at% Fe, 3 at% Cr, and

2.6at% Sc iso-concentration surfaces shown in purple, blue, and orange, respectively. **j** HADDF-STEM images and corresponding STEM-EDS results, acquired along [110]$_{Al}$ zone axis, showing Al₃(Sc, Zr) particles in the α-Al matrix. **k** The composition profiles across the interface between a-Al and Fe/Cr rich region reveal the Fe/Cr rich region to be of the stoichiometry of Al₁₃(Fe, Cr)₂.₇. Error bars represent 1σ statistical uncertainty based on atom counting in each sampling bin. **l** The composition profile of Sc (orange color) across the a-Al/ Al₁₃(Fe, Cr)₂₋₄ interface shows interfacial segregation of Sc. **m** The Zr (green) composition within the particles is both higher than in the Al matrix.

neighbor (1NN) distance. This finding suggests that vacancy trapping is not limited to single Cr atoms. Rather, stable multicomponent solute-vacancy clusters form, as confirmed by the positron annihilation lifetime measurements above (Fig. 4a).

Guided by these observations, we performed systematic DFT calculations (Supplementary Note 4) covering a series of configurations, including the monatomic-monovacancy (M-V), diatomic-monovacancy (Fe-Cr-V), triatomic-monovacancy (Fe-Cr-Sc-V), and

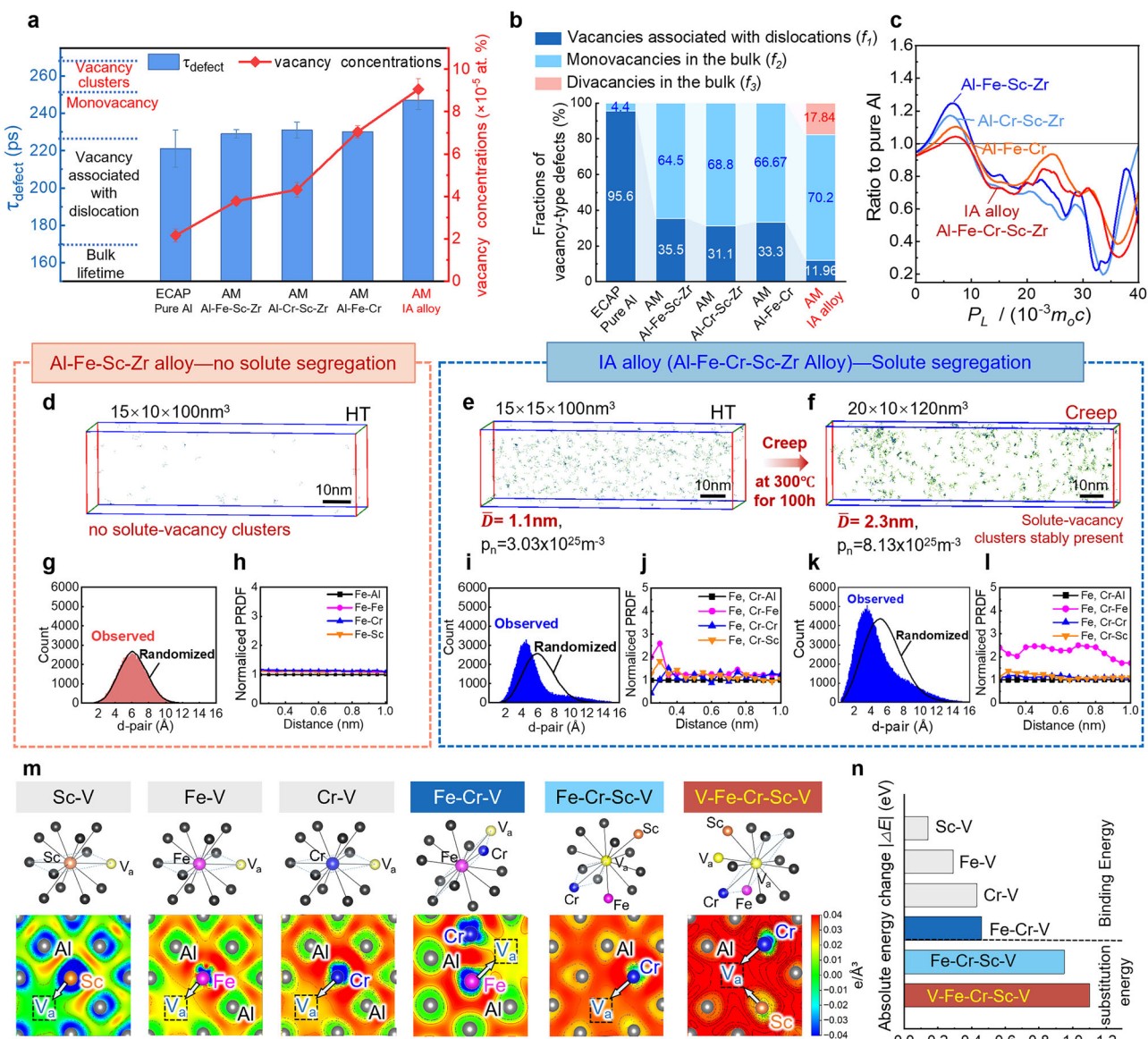

**Fig. 4 | The formation of solute-vacancy clusters in IA alloy for capturing vacancies. a** Positron annihilation lifetime and vacancy concentrations of AM Al-Fe-Sc-Zr, Al-Cr-Sc-Zr, Al-Fe-Cr, and IA alloys, and comparison with equal channel angular pressing (ECAP) pure Al[41]. Bars represent the mean ($n = 3$), individual data points are overlaid, and error bars denote the standard deviation. **b** Fractions of vacancy-type defects. **c** CDB ratio curves for AM Al-Fe-Sc-Zr, Al-Cr-Sc-Zr, Al-Fe-Cr, and IA alloys normalized to pure Al. Identifying solute-vacancy clusters using the maximum separation method: **d** Al-Fe-Sc-Zr alloy after heat treatment; **e** IA alloy after heat treatment; **f** after 300 °C creep for 100 hours. **g** Histogram of the solute 2NN pair distance; **h** Partial radial distribution functions (PRDF) with respect to the Fe solute atoms within the Al matrix in Al-Fe-Sc-Zr alloy. **i, k** Histogram of the solute 2NN pair distance; **j, l** PRDF with respect to the Fe, Cr solute atoms within the Al matrix in IA alloy after HT and Creep. **m** Deformation charge density difference maps of monatomic-monovacancy, diatomic-monovacancy (Fe-Cr-V), triatomic-monovacancy (Fe-Cr-Sc-V), and triatomic-divacancy(V-Fe-Cr-Sc-V), where red and blue represent charge accumulation and depletion, respectively. The enhanced binding energy in solute-vacancy clusters stems from the increase in charge density. **n** Binding energy and substitution energy of the lowest site occupation after relaxation for the above solute-vacancy clusters, the triatomic-divacancy (V-Fe-Cr-Sc-V) exhibits the strongest stability.

triatomic-divacancy (V-Fe-Cr-Sc-V) ones. Visualized deformation charge density difference (DCDD) maps (Fig. 4m) reveal a significant increase in charge density within the V-Fe-Cr-Sc-V system, indicating a stronger tendency for binding vacancies, namely easier formability and higher stability of such a clustering configuration. Details of the DCDD calculation are described in the Supplementary Note 5. Moreover, our calculations demonstrate a marked increase in solute-vacancy binding energy $|\Delta E|$ for Fe-Cr dual-atom systems versus single-atom systems. These binding energies, calculated based on optimal site occupation following supercell relaxation (Supplementary Fig. 11), reflect interactions with available vacancies. Furthermore, substitution energy calculations, quantifying the system energy change after

replacing one atom with another, confirm that the triatomic-divacancy configuration (V-Fe-Cr-Sc-V, where Sc acts as the substituting atom) is the most stable solute-vacancy cluster (Fig. 4n). This implies that vacancy trapping does not rely solely on the anticipated Cr atoms, but rather the synergistically co-trapping divacancies by Fe, Cr, and Sc atoms, which promotes cluster formation and stabilizes them. Such abundant small sized dispersed clusters with a size of 1~2 nm constitute a counterintuitive stabilization contrary to traditional wisdoms, necessitating two key conditions which are exactly involved in the present IA strategy: (i) high initial vacancy concentration achievable via severe deformation or rapid solidification, and (ii) strategic alloying with elements (e.g., Cr, Sc) possessing strong vacancy-binding energies

to transform detrimental free point defects into trapped nano-complexities.

The coarsening of heat-resistant particles occurred via vacancy-mediated mass transfer of precipitate forming atoms, underscoring the critical importance of controlling this mechanism. Motivated from the IA design strategy, engineered segregation generated by inherent atomic attractions are expected to enhance the thermal resistance of precipitates in our AM-Al alloy[44,45]. To investigate the precipitate sta-bilization, APT analyses was conducted on the two key precipitates, $L1_2$ nanoscale precipitates and $Al_{13}(Fe, Cr)_{2-4}$ phases. Examining the atomic-scale morphologies of Fe/Cr-modified $L1_2$ and Sc/Zr-modified $Al_{13}(Fe, Cr)_{2-4}$ phases in the samples under long-term ageing and creep deformation at 300/350°C, we aimed to clarify the coarsening mechanism of such multi-component phases exposed to high tem-peratures. As seen in Fig. 5a, dispersed $L1_2$ precipitates are maintained with an average size of ~3.86 nm and number density of ~$4.32 \times 10^{23}$ m$^{-3}$ after the ageing. After creep deformation at 300 °C and 350°C, the average size of $L1_2$ phases slightly increases to ~4.78 nm and ~5.82 nm (Fig. 5b, c), while their number densities raise up to ~$0.85 \times 10^{24}$ m$^{-3}$ and ~$1.15 \times 10^{24}$ m$^{-3}$. Illustrated by the concentration profiles of APT analyses, $L1_2$ precipitates always possess a Zr-enriched core ($C_{Zr-L12} \sim 0.5 - 1$ at%) with Sc as the main constituent (~$2 - 8$ at%) in the shell, as shown in Supplementary Fig. 12. Notably, coherent $L1_2$-$Al_3Sc$ particles (size <10 nm) substantially strengthen Al alloys due to their ability of inhi-biting dislocation movement[46]. However, once over 300°C, the ther-mally motivated disruption of Sc atomic ordering inevitably leads to a remarkable reduction in shear resistance[47], particularly under low-stress high temperature creep conditions, dislocations can easily bypass these softened precipitates through slipping or climbing. Regarding this dilemma, the IA alloy manipulates Zr to replace some Sc sites in the $Al_3Sc$ lattice, forming a species of $Al_3(Sc, Zr)$ local ordering which retains all beneficial properties of $Al_3Sc$ but has higher thermal stability. Motivated by the binding-energy calculations in Fig. 1d, Fe and Cr are attracted into the $L1_2$ precipitates, and their enrichment is further intensified during creep deformation, as shown in Supple-mentary Fig. 12. They inherently form strong bonds with the free Sc/Zr atoms migrating around the matrix-particle interface, leading to extensive trapping of Sc/Zr by the $Al_{13}(Fe, Cr)_{2-4}$ precipitates. Conse-quently, the amount of Sc/Zr incorporated into each $L1_2$ particle is reduced, yielding substantially finer and more stable $L1_2$ precipitates compared with those in conventional Al-Mg-Sc-Zr alloys[48,49].

Furthermore, $Al_{13}(Fe, Cr)_{2-4}$ particles also demonstrate unex-pectedly thermal stability, coarsening by less than 15% during creep deformation, as seen in Supplementary Fig. 13. Cross-sectional under different conditions reveals that Sc and Zr tend to segregate at the matrix-particle interfaces (Fig. 5d-f), which is attributed to the antici-pated binding effect. 2D concentration plots (Fig. 5g-i) indicate that the increased temperature progressively promotes the Sc/Zr enrichment inside the particles during creep deformation, reducing the Fe/Cr incorporation into the particle and inhibiting the coarsening[50]. Similar to conventional AM-Al alloys, the atomic-scale stabilization for heat-resistant particles in Al alloys also applies: (i) solute atoms segregate at interfaces (or inner) further lowers the interface energy, namely reducing the driving force for particle coarsening[51]; (ii) it suppresses Ostwald ripening by establishing localized chemical potential barriers[44]. Notably, the two effects may not play a leading role in our IA alloy, due to the confined volume of segregations at limited area of interfaces.

STEM examination at the atomic scale (Supplementary Fig. 14) reveals that Sc/Zr atoms do not change the crystal structure of the particles, and most Sc atoms dissolve into the $Al_{13}(Fe, Cr)_{2-4}$ particles. Unlike forming local periodic orders in Al-Cu alloys[44], Sc atoms can directly occupy some lattice sites in $Al_{13}(Fe, Cr)_{2-4}$. To model such a Sc substitution in $Al_{13}(Fe, Cr)_4$ crystals, all substitution possibilities were considered (10 substituting sites in Supplementary Fig. 15). Due to its

low content, similar calculations of Zr-involved models were not con-ducted, but it is believed that the segregation mechanism should not differ from the case of Sc. From a thermodynamic perspective (Sup-plementary Note 6), Sc substitution at Fe/Cr sites is energetically unfavorable, resulting in a higher solution energy. Conversely, Sc substitution at Al-9 sites of $Al_{13}(Fe, Cr)_4$ is the most favorable, exhi-biting the lowest solution energy (-0.33 eV), as shown in Fig. 5j. Fur-thermore, placing a second Sc atom at various Al-X (X = 1 to 8) sites within a $Al_{13}(Fe, Cr)_4$ lattice already containing a Sc at the Al-9 site, results in the relative energy (the normalized total energy for a given distribution configuration with two Sc atoms) differing by only ~ -0.1 eV (Fig. 5l, black data). This near-degeneracy implies a weak thermo-dynamic driving force for double-Sc cluster formation. However, considering three configurations where two Sc atoms co-occupy Al-9 sites (Fig. 5k), a much stronger driving force for Sc-Sc clustering is revealed when the second Sc atom is located at the nearest-neighbor site, where the relative energy is lowered substantially to -0.5 eV (i.e., the configuration A) as shown in Fig. 5l. Such a large energy reduction highlights the tendency of multiple Sc atoms occupying Al-9 sites of $Al_{13}(Fe, Cr)_4$ lattice, and promotes Sc aggregating in the particle. Notably, most Sc atoms have segregated at the interface rather than inside Al-dominant particles (Fig. 5d-f), mainly due to the limited Al-9 sites within the given particles. Besides the motivation from Al-9 site occupation, the intrinsic attraction between Fe/Cr and Sc atoms also comes into play. Lots of free Sc atoms in alloy matrices have been captured by Fe/Cr atoms to form nanoscale Fe/Cr-Sc clusters, because of their strong binding tendency. As well, the increased temperature or prolonged thermal exposure facilitates long-term atomic migration, allowing more Sc atoms to enter into the $Al_{13}(Fe, Cr)_4$ particles. Then the thermodynamically favorable Sc substitution occurs, ultimately promoting the generation of internal Sc-rich clusters beneficial for the higher thermal stability of those particles (shown in Fig. 5e, f and Supplementary Fig. 16).

In summary, this work demonstrated an Intrinsic Attraction (IA) strategy for element screening at atomic scale, which transcends the framework of traditional low-diffusivity alloying design. By leveraging the intrinsic attraction between atoms and defects, the strategy pre-engineers the segregation tendency of specific elements at defect sites, strengthens the intrinsic attraction between atoms and defects, and thereby traps them across multiple dimensions—ultimately improving the strength and thermal stability of both the matrix and precipitates. Guided by this IA strategy, we have successfully fabri-cated an exceptional heat-resistant Al alloy (referred to as the IA alloy) via PBF-LB. Such IA alloys simultaneously achieve extraordinarily high strength (UTS ≈ 530 MPa at room-temperature, 305 MPa at 300°C and 190 MPa at 400 °C), satisfactory creep resistant, and good processa-bility (porosity <0.1%, crack-free in a large-size bladed disk).

Experiments confirm the effectiveness of this preset strengthen-ing mechanism from intrinsic attraction: (i) high-concentration vacancies were anchored, promoting the formation of V-Fe-Cr-Sc-V divacancy clusters to stabilize the matrix; (ii) dislocations were cap-tured, through solute atmospheres dragging dislocation climb to decrease the steady-state creep rate; (iii) precipitating atoms were attracted, generating Sc/Zr segregations at boundaries and inside $Al_{13}(Fe, Cr)_{2-4}$ nanoprecipitates to hindering particle coarsening. This concept of enhancing strength and thermal stability through multi-scale defect control is expected to provide effective design ideas for next-generation heat-resistant structural materials used in extreme environments, and it can be readily extended to other materials for widespread applications.

## Methods
### Density functional theory (DFT) simulation
All density-functional theory (DFT) calculations were performed with the program of Vienna ab-initio Simulation Package (MedeA VASP 6).

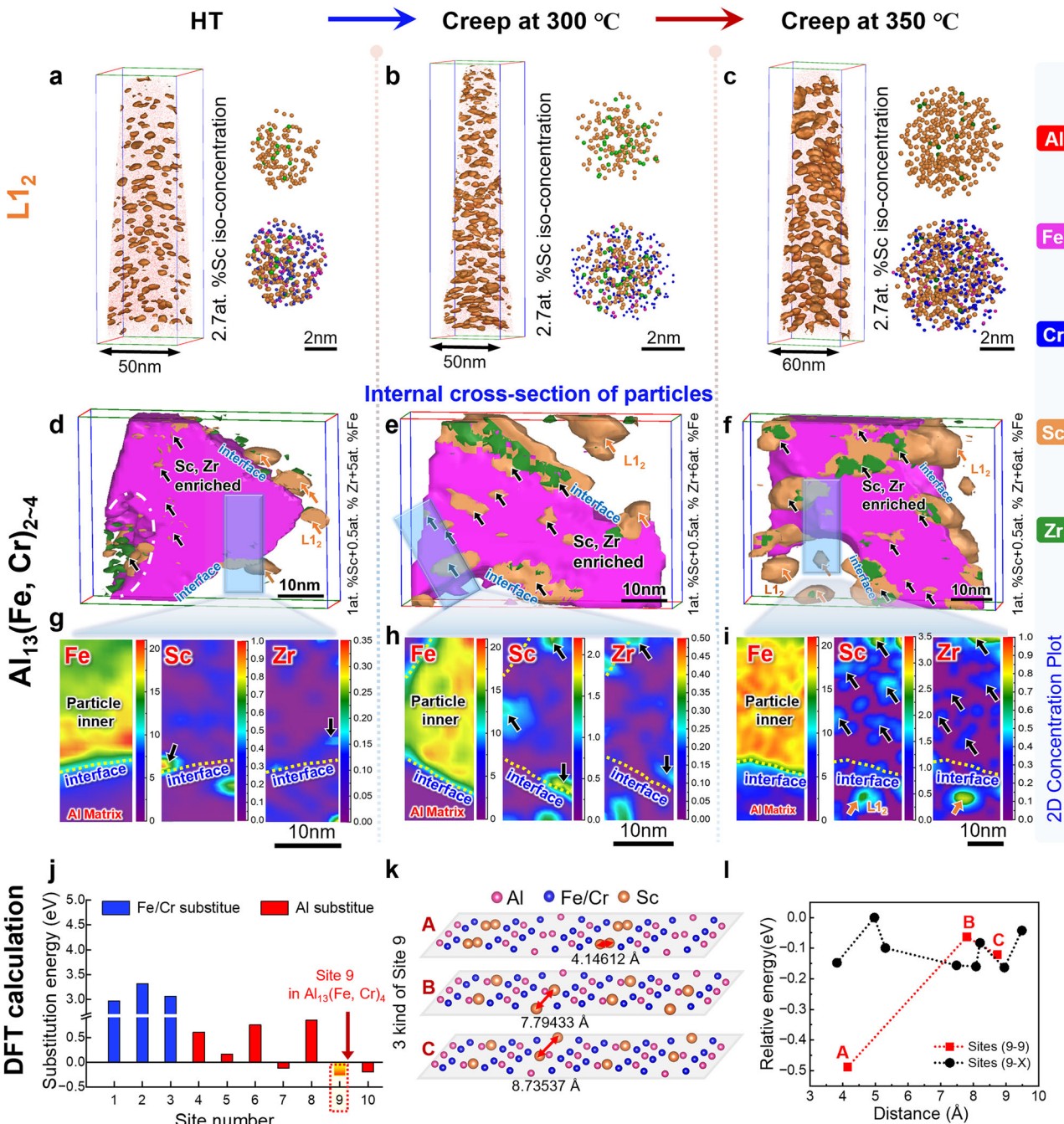

**Fig. 5 | The APT characterization of precipitates and calculated substitution energy.** 3D-APT reconstructions of **a**–**c** L1$_2$-Al$_3$(Sc, Zr) and **d**–**f** Al$_{13}$(Fe, Cr)$_{2\sim4}$ particles after HT, creep at 300 °C and creep at 350 °C, **a**–**c** illustrate the distribution, size change and elemental enrichment variation of L1$_2$ particles; **d**–**f** show the internal cross-section of Al$_{13}$(Fe, Cr)$_{2\sim4}$ particles, it found that Sc/Zr enriched inside and at the boundaries of Al$_{13}$(Fe, Cr)$_{2\sim4}$ particles, while purple for Fe atoms, orange for Sc atoms and green for Zr atoms. Black arrows indicate the Sc/Zr-enriched regions within the particles, while orange arrows highlight the L1$_2$ precipitates distributed in the matrix. **g**–**i** 2D composition analysis of Fe, Sc, Zr elements inside

Al$_{13}$(Fe, Cr)$_{2\sim4}$ particles under different states, the upper end represents the particle inner, while the lower end represents the interface between the particle and the Al-matrix. **j** The solution energy for various substitution configurations of two Sc atoms within the Al$_{13}$(Fe, Cr)$_4$ lattice calculated by DFT shows that site 9 is energetically favorable. **k** A, B, and C stand for three potential configurations resulting from Sc substitution at site 9 **l** DFT-calculated energy variation of the separation distance between two substitutional Sc atoms. The data point marked site (9-9) signifies both Sc atoms are at site 9, while site (9-X) means only one Sc atom is at site 9.

The generalized-gradient approximation (GGA) in the Perdew-Burke-Ernzerhof (PBE) form and the projector-augmented-wave (PAW) pseudopotentials[52] were employed to treat exchange–correlation and electron-ion interactions, respectively. A plane-wave cutoff energy of 400 eV was chosen, and the k-point meshes for every supercell were generated with a spacing of 0.03 Å$^{-1}$. The convergent of structural

relaxations and electronic self-consistency were deemed when the force on each atom fell below 0.02 eVÅ$^{-1}$ and the total energy difference between successive iterations was less than 1×10$^{-5}$ eV. Beyond the general computational settings described above, all DFT-specific equations, model construction procedures and key assumptions for each system are detailed in Supplementary Note1-6.

## Material processing

The raw powders with a nearly spherical morphology were gas atomized using nitrogen gas. After sieving, powders within the size range of 15-53 μm were subjected to PBF-LB fabrication. Testing samples were fabricated using a commercial BLT-A160 machine, equipped with a 400 W fiber laser at a spot size of 100 mm, under the optimized parameter set of laser power of 225 W, scan speed of 1300 mm/s, layer thickness of 0.03 mm and hatch distance of 0.08 mm. A rotation angle of 90° was used between layers. The selected parameters from the initial investigation of 30 different sets resulted in a built material with optimal density, obtaining high-dense samples with a porosity less than 0.15%. Rectangular samples with approximate dimensions of 75 × 15 × 15 mm (width × length × height) were built in a high-purity argon atmosphere with an oxygen concentration of below 500 ppm. Besides the standard rectangular coupon, a bladed disk system was also printed using the identical parameter set (Fig. 2f), confirming process robustness across geometries. The long axis is parallel to the build plate, which was heated to 100 °C to reduce residual thermal stresses. The as-fabricated samples were subjected to a stress-relief anneal at 175 °C for 2 h prior to removal from the build plate in order to further reduce residual stresses. The chemical composition of powders and PBF-LB fabricated samples were measured via inductively coupled plasma-atomic emission spectrometry (ICP-AES), the results are provided in Supplementary Table 1.

## Microstructural characterization

Microstructural characterization was conducted using a TESCAN CLARA scanning electron microscope (SEM) equipped with an Oxford Instruments Ultim Max 40 energy-dispersive spectroscopy (EDS) system, operating at an accelerating voltage of 5 kV and a working distance of 10 mm to analyze surface morphology and elemental distribution. High-resolution imaging and localized elemental mapping were performed by a Talos F200X scanning transmission electron microscope (STEM) at 200 kV, coupled with a Super-X EDS detector for nanoscale compositional analysis. Additionally, a Thermo Scientific Spectra 300 transmission electron microscope (TEM) operated at 200 kV was employed to acquire high-resolution lattice fringes and selected-area electron diffraction (SAED) patterns, with quantitative micro-area composition analysis achieved via its integrated high-sensitivity dual-pump silicon drift detector (SDD-EDS). All experiments were carried out at room temperature, with samples subjected to ultrasonic cleaning and ion milling to ensure surface cleanliness and electron transparency. The grain structure and crystal orientation were determined through Electron Back Scattering Diffraction (EBSD, Regulus8230). The samples, post electrolytic polishing at 25 V for 120 seconds, underwent examination. The EBSD data were subsequently analyzed using the Aztec Analysis software. The STEM foils (3 mm diameter) were prepared via a twin-jet electropolishing device (TJ100-SE, LEBO Science) with an electrolyte of 30% $HNO_3$ and 70% methanol, operating at 25 V and -30 °C. To identify the phase constituents in both atomized powder and deposited samples, an X-ray diffractometer (X-ray Diffraction, D/max 2500 pc) was employed with the 2θ angle range from 20° to 80°. The X-ray diffractograms were analyzed using Jade 9 software. Atom-probe tomography (APT) was performed using a CAMECA LEAP 5000XR instrument to investigate the chemistry of intermetallic phases. Nanotips were fabricated via the standard lift-out technique in a Helios Nanolab G3 UC dual beam focused ion beam-SEM. Initially, wedges were extracted from polished specimens, affixed to Si micro-posts, honed to approximately a 100 nm radius using a 25 kV $Ga^+$ beam, and polished with a 1.5 kV $Ga^+$ beam. APT analysis was conducted in laser mode with a set temperature of 50 K, operating in a high vacuum environment under a pressure below $2.0 \times 10^{-9}$ Pa ($1.5 \times 10^{-11}$ Torr). The laser kept a pulse repetition rate of 250 kHz and the energy was calibrated for each individual tip to yield an equivalent pulse fraction in voltage mode of 20%, which corresponded to laser energy between 75 and 160 pJ. The detection rate was set to have 0.5% of the applied laser pulses resulting in an evaporation event. Data reconstruction and analysis were performed using AP Suite 6.1 software.

Partial radial distribution function (PRDF) analysis was conducted to process LEAP tomographic data, enabling quantitative assessment of solute-cluster characteristics. PRDF was used to resolve the local spatial distribution of solute atoms: spherical shells centered on either solute atoms or vacancies were defined, the ion density within each shell was counted, and the resulting radial profiles for different chemical environments were calculated. The peak profiles including position, width, and integrated intensity were analyzed from Gaussian fitting, and then key parameters of average cluster size, nearest-neighbor coordination number, and interfacial roughness were extracted.

Solute clusters and precipitates were identified via APT reconstructions in a well-established maximum-separation method[43], to gain the critical parameters, e.g., the maximum separation distance between adjacent solute atoms within a cluster, $d_{max}$, and the minimum number of atoms required to define a cluster, $N_{min}$. Among them, $d_{max}$ was determined as the distance at which the cumulative probability difference in first-nearest-neighbor distances between experimental and randomized datasets reached its maximum. $N_{min}$ was established by comparing the size distributions of clusters detected in experimental data with those generated from randomized solute distributions. The final parameters were set to $d_{max} = 9$ Å, $N_{min} = 12$, $L = 6$ Å, and $D_{erosion} = 6$ Å.

## Positron annihilation

Positron annihilation lifetime spectroscopy (PALS) was employed to characterize vacancy-type defects and their concentrations in the alloys. In this technique, injected positrons are trapped at open-volume defects such as vacancies, vacancy clusters or dislocations, where the reduced electron density leads to a longer positron lifetime compared with that in the defect-free lattice. Analysis of the lifetime components therefore provides information on the type and relative concentration of defects in the material[41]. AM Al-Fe-Sc-Zr, AM Al-Cr-Sc-Zr, AM Al-Fe-Cr and AM IA samples were cut into ~10 mm×10 mm×1 mm slices. Positron lifetime measurements were conducted in nitrogen atmosphere using a DPLS-4000 digital spectrometer based on an ORTEC fast-fast coincidence system. The time resolution of the system is about 190 ps in full width at half maximum (FWHM). A 1.0 MBq $^{22}$Na source, sealed between two 7.5 μm Kapton foils, was sandwiched between identical sample pairs. The 1.28 MeV γ photon provided the start signal, and the 0.511 MeV annihilation γ photon the stop signal. Each spectrum contained å $1.5 \times 10^6$ counts and was analyzed with the Lifetime-9 program to extract lifetime components and their intensities, thereby characterizing vacancy-type defects. For additional details of the positron annihilation techniques, see Supplementary Note. 7.

## Coincidence Doppler Broadening Spectroscopy

Coincidence Doppler broadening spectroscopy (CDBS) was conducted with a DCDB-3000 digital spectrometer equipped with two HPGe detectors (energy resolution≤1.3 keV at 511 keV, relative efficiency≈35%). Spectra were collected for ~12 h, accumulating≥$5 \times 10^6$ counts. Meanwhile, the Doppler broadening spectrum of the standard sample Al was measured as the reference spectrum. Figure 4c shows the CDB ratio curves, where the error bars are standard deviations of the mean. $P_L$ is the longitudinal component of the positron electron momentum along the direction of the γ-ray emission, $c$ is the speed of light, $m_O$ is the electron rest mass. For additional details, see Supplementary Note. 7.

## Mechanical testing

Process a tensile specimen with a diameter of 5 mm and a gauge length of 30 mm. Room-temperature uniaxial tensile tests were performed

using Instron 3369 electronic universal material testing machine, with a strain rate of $10^{-3}$ mm/s. High temperature tensile tests were conducted on a hydraulic servo dynamic testing machine (8802, UK), with a $10^{-4}$ s$^{-1}$ initial strain-rate, which was increased to $10^{-3}$s$^{-1}$ at 1.5% strain. In order to ensure uniform temperature distribution of specimens during mechanical testing, all tensile specimens are heated to the target temperature and held for 20 minutes before loading. Three identical replicates were tested at each temperature. Tensile samples for creep tests were of the same dimensions as the tensile specimens mentioned above. Tensile creep tests were performed using Sust electrical Equipment (RMT-D5G) with a maximum load of 50 kN. The temperature was measured with three evenly tied thermocouples along the sample gauge length, while the exact elongation was obtained from a dual-gauge extensometer positioned on the gauge section of the samples.

## Data availability

Relevant data supporting the key findings of this study are available within the article and the Supplementary Information file. The source data generated in this study have been deposited in the Figshare repository under the accession code Doi: 10.6084/m9.figshare.31403481.

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

## Acknowledgements

This work was supported by the Fundamental and Interdisciplinary Disciplines Breakthrough Plan of the Ministry of Education of China (JYB2025XDXM409, R. L.); National Natural Science Foundation of China (Grant numbers U21B2073, R.L.; 52571056, K.G.); Guizhou Provincial Science and Technology Projects (No. [2025]044, R.L.).

## Author contributions

R.L. proposed and supervised the project. Y.W. performed most of the properties and the characterizations. C.Y. performed the DFT calculations. Y.W., C.Y., K.G., T.Y. and R.L. analyzed the data. Y.W., C.Y., K.G., T.Y. and R.L. contributed to the result discussion. Y.W. and C.Y. prepared samples. Y.W., K.G. and R.L. conceptualized the manuscript. The final version was approved by all authors before submission.

## Competing interests

The authors declare no competing interests.
