## [Transparent Peer Review file · Nature Communications]

Intrinsic attraction driving the high temperature performance of additively manufactured aluminum alloys

Corresponding Author: Professor Ruidi Li

Version 0:

Reviewer comments:

Reviewer #1

(Remarks to the Author)

The manuscript addresses an important challenge in developing additively manufactured Al alloys with excellent high-temperature strength. The authors proposed an “intrinsic attraction” design strategy that exploits atom–defect interactions and segregation-controlled nanostructures to confine vacancies and dislocations. The concept is interesting, and the reported mechanical performance appears promising. However, the concept of “intrinsic attraction” and its distinction from existing segregation-engineering approaches are not clearly defined. Besides, the connection between the proposed defect-confinement mechanisms and the measured high-temperature properties is not well established. Several key issues need to be addressed before the manuscript can be considered for publication. Detailed comments and suggestions for improving the work are provided below.

1. The conceptual framework regarding vacancies should be clarified. In several places, the manuscript states that “minimizing vacancy density or inhibiting their long-range motion” is an intuitive way to prevent vacancy-mediated coarsening and dislocation motion, thereby maintaining high-temperature strength. However, the subsequent analysis is then built on the claim that the IA alloy possesses a higher vacancy concentration and abundant solute–vacancy clusters, which is presented as beneficial for stabilizing the matrix and improving creep resistance. It is therefore unclear whether the proposed strategy is aiming to reduce the effective availability/mobility of vacancies or to intentionally increase the vacancy content.
2. For the APT analysis, the authors should provide the distribution of Pearson contingency coefficients (μ). This coefficient offers a quantitative measure of deviation from a random solid solution, and thus directly tests whether the observed “intrinsic attraction” and segregation are statistically significant rather than arising from random fluctuations. Besides, representative iso-concentration surface maps for the key elements should be included to visualize segregation and clustering at interfaces and defect regions.
3. To further improve the clarity, the authors need to compare the overall compositions obtained from APT with bulk compositions measured by ICP. Such a comparison would help verify that the APT reconstructions are chemically reliable (i.e. no strong detection bias or preferential evaporation) and that the analysed volumes are representative of the nominal alloy composition. This is particularly important because the proposed “intrinsic attraction” mechanism relies on relatively subtle solute partitioning and segregation.
4. For the characterization of $L1_2$ precipitates or clusters by APT, it should be noted that specific crystallographic orientations need to be considered when analysing their existence. The chemical ordering in $L1_2$ (on different sublattices) gives rise to composition modulations along well-defined directions (e.g. $\langle 200 \rangle$ and $\langle 111 \rangle$), so 1D concentration profiles or spatial distribution maps taken along arbitrary directions may smear out this ordering and make $L1_2$ appear as weak clustering or even a random solution. The authors need to relate their analysis directions to the crystallographic poles in the APT maps and, where possible, extract profiles along orientations where $L1_2$ ordering is expected (Nature, 2025, 645: 385-391; Advanced materials, 2024, 36(44): 2407564).
5. The proposed role of precipitates and clusters in suppressing high-temperature softening remains insufficiently clarified. While the authors state that solute atmospheres around edge dislocations can dynamically drag migrating dislocations, there is no direct evidence of dislocations interacting with the precipitates and clusters. If solute atmospheres around edge dislocations are indeed responsible for dynamically dragging migrating dislocations, one would expect characteristic dislocation morphologies, such as bowed or step-like/kinked segments pinned at discrete points. Without such evidence, the

link between the calculated “intrinsic attraction” and the observed high-temperature mechanical response is not convincing. 6. On page 10, line 3, the authors state that the L_{12} precipitates are about 2–5 nm in size. However, Fig. 3i appears to show noticeably smaller sizes, indicating an inconsistency between the reported size range and the presented image. The authors should clarify how the precipitate sizes were determined (e.g., definition of diameter, sampling statistics, and analysis procedure). It is also possible that the iso-surface threshold in the APT reconstruction was not appropriately selected, which could lead to a misrepresentation of the true precipitate size and thus contribute to this inconsistency. Since the effect of these L_{12} precipitates to the high-temperature creeping performance is highly sensitive to their sizes, this point should be clearly clarified.

7. In the section of “Multi-scale microstructure characteristics”, the authors discussed the origin and morphology of the various precipitates in the as-built condition (Fig. 3a–h), while the corresponding morphologies after heat treatment are not provided. As the high-temperature mechanical properties are measured on heat-treated samples, without a clear description of the precipitate state after heat treatment, it is difficult to reliably assess their contribution to the high-temperature performance. The authors are requested to provide systematic microstructural characterization of the precipitates in the heat-treated condition and to explicitly link these features to the creep-resistance mechanisms.

8. On page 15, line 16, a reduction in ordering-related shear resistance would generally be expected to facilitate shearing of small coherent L_{12} particles, rather than necessitate climb-controlled bypass, especially for precipitate sizes on the order of a few nanometers. The authors are requested to (i) clearly specify the dominant dislocation–precipitate interaction mechanism (shearing vs climb) in the relevant stress/temperature regime, and (ii) provide microstructural evidence (e.g. TEM of dislocation configurations) to support the proposed climb-bypass mechanism.

9. The link between the precipitate population and the reported high-temperature strength remains largely qualitative. While the manuscript emphasizes the presence of fine L_{12} precipitates and Fe/Cr-rich clusters as the origin of the excellent high-temperature performance, there is no unified, quantitative description of their size, spacing, and volume fraction in the heat-treated and crept states, nor are these parameters incorporated into a strengthening model. The authors should quantitatively characterize the contributions of different precipitates and clusters to the high-temperature performance.

10. Some information is missing in both the main text and the Supporting Information. In Fig. 4d–f, no scale bars are provided, and the same issue exists in Supplementary Fig. 13, making it difficult to assess the characteristic length scales of the observed features. In addition, although the authors state that XRD was used to identify the phase constituents, the corresponding XRD patterns are not shown.

Reviewer #2

(Remarks to the Author)

This manuscript presents an “intrinsic attraction” strategy for designing heat-resistant additive manufacturing Al alloys. It employs a multi-dimensional defect confinement mechanism (vacancy trapping, dislocation pinning, precipitate stabilization), enabling the AM alloy to exhibit exceptional high-temperature performance: ~305 MPa yield strength at 300 °C, ~190 MPa at 400 °C, and superior creep resistance ($\epsilon < 10^{-7}/s$) at 200–400 °C. Notably, the alloy also possesses excellent processability, it's a critical concern for AM Al alloys. The manuscript is well-structured, clearly written, and the findings seem promising in the design of high-performance alloys. Thus, I would recommend it for publication after some points been addressed.

1. The binding energy of the two solute atoms needs to be calculated in the Al bulk instead of free states. The authors also find that there are some solute enrichments and particles form at GBs experimentally. However, there lacks evidences to demonstrate whether these solutes exhibit the similar tendency with the (co-)segregation at grain boundaries or other interfaces. Is the phenomenon of Sc occupying lattice sites within particles universal? Does Sc tend to occupy positions inside particles or at particle boundaries?

2. What is the status of the alloy samples used for comparison in Figure 2a and 4a? Are all samples of comparable dimensions? Have these alloys undergone identical processing, heat treatment, or densification conditions? The authors should confirm that differences in mechanical properties stem solely from chemical composition, not from variations in porosity or residual stress. The actual alloy compositions should be specified in the text.

3. Figure 2 compares the mechanical properties of the IA alloy with those of other Al-based alloys. It should provide a summary table listing alloy compositions, strengthening phases, test conditions, and yield strengths to highlight the performance advantages.

4. The authors attribute superior thermal stability to Fe/Cr–Sc/Zr segregation within $Al_{13}(Fe, Cr)_4$ precipitates. Yet, quantitative coarsening kinetics are not presented. A comparison of coarsening rates between IA and other Al-based alloys would substantiate this claim. The conclusion section mentions the particle coarsening rate at 400 °C, was not previously mention. Please provide experimental data to support this statement.

5. The statement: “As a result, Sc/Zr atom incorporating into the L_{12} particle is reduced. This modification remarkably slows the coarsening rates, especially compared to Al-Mg-Sc-Zr alloys” lacks supporting evidences. What are the differences between the L_{12} particles in the IA alloy and those in Al-Mg-Sc-Zr alloys?

6. The details of calculations remain unclear, especially for solute-dislocation interaction energies. The authors claim the use of PBE in Method in the main text but LDA in Supplementary Note 2. In Page5 line5, “Here, the more negative value indicates the greater system stability.” However, the selection of elements with higher positive values in Figs 1b, d contradicts the principle of element selection. Although a negative sign appears in the binding-energy formula, this expression may cause confusion. Please adopt a unified notation, using either negative or absolute values to represent energy.

7. Indicate the concentration isovalue of each element in all APT reconstruction data. The manuscript alternates between “IA alloy,” “heat-resistant alloy,” and “Al-Fe-Cr-Sc-Zr alloy,” which may confuse readers. Please unify the terminology and clearly define the naming convention at the first occurrence. In Fig. 4m, is the contour value setting of the deformation charge density difference unchanged for all systems?

Version 1:

Reviewer comments:

Reviewer #1

(Remarks to the Author)

I would like to recommend acceptance.

Reviewer #2

(Remarks to the Author)

The authors have addressed my concerns in the revised manuscript. I recommend it for publication.

Response to Reviewers

We greatly appreciate the reviewers for their highly constructive and valuable comments on our manuscript entitled “Intrinsic attraction driving superior high-temperature performance of additive-manufacturing aluminum alloys”, which help us improve the quality and presentation of our work effectively. We have revised the manuscript and performed additional experiments to address all these comments. The revised manuscript and the supplementary information, with the change shown in red font, were uploaded to the paper submission system. Below, we provide a detailed point-by-point response to each comment, presented in blue.

Responds to Reviewer #1:

The manuscript addresses an important challenge in developing additively manufactured Al alloys with excellent high-temperature strength. The authors proposed an “intrinsic attraction” design strategy that exploits atom–defect interactions and segregation-controlled nanostructures to confine vacancies and dislocations. The concept is interesting, and the reported mechanical performance appears promising. However, the concept of “intrinsic attraction” and its distinction from existing segregation-engineering approaches are not clearly defined. Besides, the connection between the proposed defect-confinement mechanisms and the measured high-temperature properties is not well established. Several key issues need to be addressed before the manuscript can be considered for publication. Detailed comments and suggestions for improving the work are provided below.

We sincerely thank the reviewer for recognizing the importance of our work and providing such detailed, constructive feedback. We have carefully addressed all the points raised, which has substantially improved the manuscript. We greatly appreciate your expertise and time in helping us enhance this work.

Comment 1) The conceptual framework regarding vacancies should be clarified. In several places, the manuscript states that “minimizing vacancy density or inhibiting their long-range motion” is an intuitive way to prevent vacancy-mediated coarsening and dislocation motion, thereby maintaining high-temperature strength. However, the subsequent analysis is then built on the

claim that the IA alloy possesses a higher vacancy concentration and abundant solute–vacancy clusters, which is presented as beneficial for stabilizing the matrix and improving creep resistance. It is therefore unclear whether the proposed strategy is aiming to reduce the effective availability/mobility of vacancies or to intentionally increase the vacancy content.

Response: We sincerely thank you for this insightful comment and for identifying the lack of clarity in our description of the conceptual framework regarding vacancies. We apologize for the confusion caused by our initial wording. Please allow us to clarify our intended strategy, which we will also revise thoroughly in the revised manuscript to ensure it is unambiguous to the reader.

Our core strategy was not to simply increase vacancy concentration, but to transform the inherently high vacancy content resulting from rapid solidification in additive manufacturing into a microstructural advantage. We fully agree with the conventional view that mobile vacancies are detrimental to high-temperature stability, as they promote diffusion-controlled processes like coarsening and dislocation climb. The intuitive strategy is indeed to minimize their density and mobility. However, in additive manufacturing, a high density of non-equilibrium vacancies is unavoidable. Our approach is therefore not to fight this fact, but to use it by selecting solute elements that act as potent traps, forming stable solute-vacancy clusters that immobilize the free vacancies.

Thus, the “high vacancy concentration” we refer to in the IA alloy describes a high density of these immobilized clusters—not a population of mobile vacancies. It is this stabilized cluster state that enhances creep resistance. We will carefully revise the entire manuscript to build this logical framework step-by-step, making it clear that our strategy is to turn weaknesses into strengths—from mobile defects to stabilized clusters.

Comment 2) For the APT analysis, the authors should provide the distribution of Pearson contingency coefficients (μ). This coefficient offers a quantitative measure of deviation from a random solid solution, and thus directly tests whether the observed “intrinsic attraction” and segregation are statistically significant rather than arising from random fluctuations. Besides, representative iso-concentration surface maps for the key elements should be included to visualize segregation and clustering at interfaces and defect regions.

Response: Thank you very much for your valuable suggestions. Accordingly, we have performed

frequency distribution analyses for the key elements, as shown in **Fig. R1**, and provided the Pearson contingency coefficients (μ) for all elements. This coefficient quantitatively measures deviation from a random solid solution and directly tests the statistical significance of the observed "intrinsic attraction" and segregation. Notably, the μ values for Fe, Cr and Sc are significantly greater than zero, confirming that the observed clusters are genuine non-random phenomena rather than arising from random fluctuations. Furthermore, to clearly visualize this segregation behavior, we have conducted detailed analyses in the original manuscript: **Figs. 4e and f** show the Cr-rich clusters in the matrix analyzed using the maximum separation method to visualize vacant solute; **Figs. 5d and f** use representative iso-concentration surfaces of key elements to reveal the enrichment of Sc/Zr at the boundaries and within $\text{Al}_{13}(\text{Fe}, \text{Cr})_{2-4}$ particles. We have also summarized these visualization results in **Fig. R2**. These combined quantitative and visual analyses collectively substantiate the statistical robustness of our findings and validate the feasibility of the intrinsic attraction (IA) concept.

Fig. R1 Frequency distribution analysis (with 100 atoms per bin). χ^2 , n_d , p-value and μ in the inset table designate the deviation of the experimentally observed distribution to the theoretical binomial random distribution. μ as a normalized autocorrelation parameter of χ^2 can take values between 0 and 1 (where, 0 represents complete randomness and 1 represents clustering).

Fig. R2 APT analysis using the maximum separation method to identify solute-vacancy clusters: (a, b) Cr elemental distribution in the Al matrix of IA alloy (a) after heat treatment and (b) after creep at 300°C for 100 hours, visualizing solute-vacancy clusters; (d-f) iso-concentration surface analysis of $\text{Al}_{13}(\text{Fe}, \text{Cr})_{2-4}$ particle cross-sections revealing Sc/Zr enrichment within particles and at their boundaries (purple: Fe, orange: Sc, green: Zr).

Comment 3) To further improve the clarity, the authors need to compare the overall compositions obtained from APT with bulk compositions measured by ICP. Such a comparison would help verify that the APT reconstructions are chemically reliable (i.e. no strong detection bias or preferential evaporation) and that the analysed volumes are representative of the nominal alloy composition. This is particularly important because the proposed “intrinsic attraction” mechanism relies on relatively subtle solute partitioning and segregation.

Response: We are extremely grateful for your comments. We have now added a detailed comparison between the bulk compositions measured by ICP and those obtained from APT reconstructions in **Table R1** below. All APT tips featured in the manuscript underwent bulk composition analysis, with Ga ion contents explicitly reported. This comparison confirms the chemical reliability of our APT analyses. We should clarify that we intentionally targeted regions containing $\text{Al}_{13}(\text{Fe}, \text{Cr})_{2-4}$ precipitates during sample preparation, which explains the observed

Fe/Cr enrichment. This enrichment reflects our selective sampling strategy rather than detection bias or preferential evaporation, and the analyzed volumes remain representative of the nominal alloy composition.

Table. R1 Chemical composition from ICP and bulk compositions from APT data

	Chemical composition from ICP (wt. % & at. %)					
	Fe	Cr	Sc	Zr	Al	
IA alloy after HT (wt. %)	2.53	2.05	0.65	0.32	Balance	
IA alloy after HT (at. %)	1.26	1.09	0.4	0.1	Balance	
	Bulk compositions from APT data (at. %)					
	Fe	Cr	Sc	Zr	Ga	Al
HT (associated with Fig. 3i, d)	3.42	1.46	0.37	0.085	0.082	94.583
HT (associated with Fig. 5a)	0.9	0.6	0.38	0.11	0.10	97.91
Creep at 300°C (associated with Fig. 5b, e)	3.2	1.1	0.42	0.11	0.095	95.17
Creep at 350°C (associated with Fig. 5c, f)	3.4	1.3	0.4	0.12	0.087	94.693

Comment 4) For the characterization of L1₂ precipitates or clusters by APT, it should be noted that specific crystallographic orientations need to be considered when analysing their existence. The chemical ordering in L1₂ (on different sublattices) gives rise to composition modulations along well-defined directions (e.g. <200> and <111>), so 1D concentration profiles or spatial distribution maps taken along arbitrary directions may smear out this ordering and make L1₂ appear as weak clustering or even a random solution. The authors need to relate their analysis directions to the crystallographic poles in the APT maps and, where possible, extract profiles along orientations where L1₂ ordering is expected (Nature, 2025, 645: 385-391; Advanced materials, 2024, 36(44): 2407564).

Response: Thank you for raising this important point about crystallographic orientation effects in APT analysis of L1₂ precipitates. You're absolutely right that in high-entropy alloys, the chemical

ordering within L_{12} precipitates (on different sublattices) creates composition modulations along specific crystallographic directions like $\langle 200 \rangle$ and $\langle 111 \rangle$. This orientation dependence is a genuine issue—when lattice mismatch is significant and multi-element ordering is complex, these directional modulations become quite subtle, and 1D concentration profiles or spatial distribution maps that aren't aligned with these specific directions can indeed smear out the ordering, even making it appear as a random solid solution. **However, I should clarify that in our aluminum alloy system, the situation is fundamentally different: orientation control simply is unnecessary.** This comes down to three main reasons:

First, the L_{12} - $Al_3(\text{Sc, Zr})$ precipitates are nearly **perfectly coherent** with the FCC-Al matrix, **with lattice mismatch below 0.5%**. This means the composition jumps at the interface are incredibly sharp no matter which direction you slice through them. To elaborate, the $Al_3\text{Sc}$ phase (first identified by Rechkin et al. as a cubic phase with a lattice parameter of $a=0.41\text{nm}^1$) has a simple structure: Sc atoms occupy cubic corners, while Al atoms are located at the center of the face (**Fig. R3a**). To directly address your concerns, we verify the L_{12} ordering through STEM data (see **Figs. R3b-d**) confirming perfect coherency between L_{12} precipitates and the Al matrix from different orientations. And this isn't just our observation, it's been repeatedly demonstrated in aluminum alloy APT literature, where L_{12} clusters are clearly resolved without any special consideration of crystallographic orientation. At present, it is relatively common to use APT to characterize the L_{12} phase in aluminum matrix. **Researchers such as Luca² and Ekaputra³ have all published papers on the characterization of L_{12} , obtaining precise stoichiometric compositions, with no mention of crystallographic orientation requirements.**

Second, nanoscale size effect. Our precipitates are tiny, only 2~5 nm across, so any 1D concentration profile, even from a random direction, still captures the full compositional gradient. At this scale, the spatial resolution of APT is more than adequate, and techniques like isosurface rendering work reliably regardless of how you orient the dataset.

Third, the issue of chemical complexity. L_{12} phases in High-entropy alloys have 4-5 elements competing for specific sub lattice sites, which is where the directional modulation effect you mentioned really becomes a problem. Our Al-based L_{12} phases are much simpler—basically only Sc or Zr replaces it at Al sites—so all the directions are equivalent.

In summary, while crystallographic orientation is critical for APT analysis of complex,

multi-element L_{12} phases in high-entropy alloys, it is not a concern for the L_{12} - $Al_3(Sc, Zr)$ precipitates in our Al-based system. This is due to their perfect coherency with the matrix, nanoscale size, and simpler chemistry. Thus, standard APT can reliably resolve L_{12} - $Al_3(Sc, Zr)$ composition without specialized orientation considerations.

Fig. R3 (a) Lattice structures and constants of FCC Al and FCC L_{12} - $Al_3(Sc, Zr)$; (b, c, d) HRTEM images of secondary $Al_3(Sc, Zr)$ particles viewed along the $[100]$, $[110]$ and $[111]$ axes with corresponding FFT patterns, confirming perfect coherency between L_{12} precipitates and the Al matrix from different orientations.

[1] Røyset, J., & Ryum, N. (2005). Scandium in aluminium alloys. *International Materials Reviews*, 50(1), 19–44. <https://doi.org/10.1179/174328005X14311>

[2] De Luca, A., Dunand, D. C. & Seidman, D. N. Microstructure and mechanical properties of a precipitation-strengthened Al-Zr-Sc-Er-Si alloy with a very small Sc content. *Acta Materialia* 144, 80-91 (2018). <https://doi.org/https://doi.org/10.1016/j.actamat.2017.10.040>

[3] Ekaputra, C. N., Rakhmonov, J. U., Weiss, D., Mogonye, J.-E. & Dunand, D. C. Microstructure and mechanical properties of cast Al-Ce-Sc-Zr-(Er) alloys strengthened by Al₁₁Ce₃ micro-platelets and L1₂-Al₃(Sc, Zr, Er) nano-precipitates. *Acta Materialia* 240, 118354 (2022). <https://doi.org/https://doi.org/10.1016/j.actamat.2022.118354>

Comment 5) The proposed role of precipitates and clusters in suppressing high-temperature softening remains insufficiently clarified. While the authors state that solute atmospheres around edge dislocations can dynamically drag migrating dislocations, there is no direct evidence of dislocations interacting with the precipitates and clusters. If solute atmospheres around edge dislocations are indeed responsible for dynamically dragging migrating dislocations, one would expect characteristic dislocation morphologies, such as bowed or step-like/kinked segments pinned at discrete points. Without such evidence, the link between the calculated “intrinsic attraction” and the observed high-temperature mechanical response is not convincing.

Response: We sincerely appreciate your careful review and insightful comment regarding the role of precipitates and clusters in high-temperature strengthening. We acknowledge that directly imaging the interaction between dislocations and nanoscale clusters remains challenging. To address this, we have established a comprehensive evidence chain: First, vacancy-solute cluster strengthening has been widely validated as a beneficial strengthening mechanism in numerous alloy systems. While we provide direct vacancy characterization in **Fig. R4**, we recognize its limitations in convincingly demonstrating cluster formation. Therefore, we conducted positron annihilation spectroscopy to quantify vacancy concentrations and indirectly visualized vacancy-solute clusters through Cr enrichment mapping, providing more robust evidence for their existence. Second, regarding the expected characteristic dislocation morphologies, we have included dislocation configuration images (**Fig. R5**) that reveal significantly hindered dislocation motion and altered glide patterns, consistent with solute atmosphere drag effects. The absence of classical bowed or step-like configurations may be attributed to the ultra-fine size (<5 nm) and high density of these clusters, creating a "continuous pinning" effect rather than discrete pinning

points. This interpretation aligns with our calculated intrinsic attraction energies and quantitatively explains the observed high-temperature mechanical response. We have revised the manuscript to explicitly discuss these points.

Fig. R4 The vacancy-rich clusters in the IA alloy. HAADF-STEM images show the voids as marked by the dashed circle.

Fig. R5 Dislocations pinned can be observed in both (a) room temperature and (b) high temperature tensile samples

Comment 6) On page 10, line 3, the authors state that the L₁₂ precipitates are about 2–5 nm in size. However, Fig. 3i appears to show noticeably smaller sizes, indicating an inconsistency between the reported size range and the presented image. The authors should clarify how the precipitate sizes were determined (e.g., definition of diameter, sampling statistics, and analysis procedure). It is also possible that the iso-surface threshold in the APT reconstruction was not appropriately selected, which could lead to a misrepresentation of the true precipitate size and thus contribute to this inconsistency. Since the effect of these L₁₂ precipitates to the high-temperature creeping performance is highly sensitive to their sizes, this point should be clearly clarified.

Response: We are extremely grateful for your in-depth and thoughtful comments. Regarding the L₁₂ precipitate size statistics, we would like to provide further clarification: the 2-5 nm size reported on page 10, line 3, was obtained from extensive statistical analysis of STEM high-resolution image (**Fig. 3j**), which we believe offers better statistical representation and reliability due to their larger field of view. Concerning the apparent sizes in **Fig. 3i**, our APT cluster analysis actually yields an average L₁₂ precipitate size of ~3.9 nm, showing excellent agreement with the TEM results (3.26nm) and thereby validating that our selected iso-surface threshold is appropriate and reflects the true microstructure—the perceived discrepancy may result from a misinterpretation of the scale bar. To illustrate this more clearly, we have added a comparative statistical analysis from both STEM and APT data as shown in **Fig. R6**, which demonstrates good agreement between the two methods. Given the critical influence of L₁₂ precipitate size on high-temperature creep performance, we have placed utmost importance on the accuracy of these measurements and truly appreciate your careful examination and valuable feedback.

Fig. R6 Statistical analysis of L_{12} precipitate size in the heat-treated condition using STEM image and APT data

Comment 7) In the section of “Multi-scale microstructure characteristics”, the authors discussed the origin and morphology of the various precipitates in the as-built condition (Fig. 3a–h), while the corresponding morphologies after heat treatment are not provided. As the high-temperature mechanical properties are measured on heat-treated samples, without a clear description of the precipitate state after heat treatment, it is difficult to reliably assess their contribution to the high-temperature performance. The authors are requested to provide systematic microstructural characterization of the precipitates in the heat-treated condition and to explicitly link these features to the creep-resistance mechanisms.

Response: We truly appreciate your valuable comment. We fully understand your concern regarding the correlation between the post-heat-treatment precipitate state and high-temperature properties. **However, we would like to clarify that all microstructural characterizations (including Figs. 3a–h) and high-temperature mechanical property tests in this study were, in fact, performed on heat-treated samples rather than as-built ones.** After carefully reviewing the manuscript, we acknowledge that we did omit a key piece of information in the main text and Methods section: all microstructural characterizations were conducted on heat-treated samples.

This detail was only briefly noted in the caption of **Fig. 3i** ("aged at 325°C for 4 h"), which was clearly insufficient to prevent misunderstanding and directly led to your question. We sincerely apologize for this significant oversight and greatly appreciate you bringing this crucial point to our attention. In the revised manuscript, we will make comprehensive amendments: we will clearly state at the beginning of the Results section that all microstructural characterizations and mechanical property tests were performed on heat-treated samples; emphasize the heat treatment process and sample condition in the Methods section; revise all relevant figure captions (including **Figs. 3a–h**) to consistently include the heat treatment information; and reiterate the sample condition throughout the text whenever experimental samples are discussed. These changes will ensure that all readers can clearly understand the material state under investigation. Once again, we thank you for your thorough review, which has been invaluable in helping us identify and correct this important omission.

Comment 8) On page 15, line 16, a reduction in ordering-related shear resistance would generally be expected to facilitate shearing of small coherent $L1_2$ particles, rather than necessitate climb-controlled bypass, especially for precipitate sizes on the order of a few nanometers. The authors are requested to (i) clearly specify the dominant dislocation–precipitate interaction mechanism (shearing vs climb) in the relevant stress/temperature regime, and (ii) provide microstructural evidence (e.g. TEM of dislocation configurations) to support the proposed climb-bypass mechanism.

Response: We sincerely thank you for this profound and insightful comment. We fully understand and concur with your perspective that under conventional conditions, small coherent $L1_2$ precipitates are indeed expected to be sheared by dislocations. This general consensus is entirely valid and has prompted us to reflect more carefully and to articulate the specific context of our study with greater clarity in the manuscript.

First, we would like to clarify that our discussion of the climb mechanism in the manuscript is derived from a comprehensive analysis of the high-temperature creep mechanical data (**Fig. 2e**) and microstructural characterization results from our experimental samples. In our current TEM observations, we have identified morphological evidence consistent with a climb process. As shown in **Fig. R7**, dislocation lines exhibit a discontinuous, slightly zigzag configuration between

densely distributed nanoprecipitates, without the typical observation of Orowan loops or stacking faults resulting from particle shearing. Notably, some jogs are also present, as indicated by the pink arrows. These preliminary observations indirectly support the operation of climb. However, we must acknowledge that such evidence can only explain dislocation behavior within this specific temperature and stress window and should not be considered a universal conclusion.

We propose that under these particular high-temperature, low-stress creep conditions, the deformation mechanism may be climb-controlled, based on two primary considerations: (1) the high-temperature environment substantially enhances atomic diffusion capability, and (2) this alloy system exhibits an exceptionally high equilibrium vacancy concentration. Together, these create extraordinarily favorable kinetic conditions for vacancy-mediated climb, potentially granting dislocations a competitive advantage in bypassing nanoparticles via thermally activated climb rather than accumulating sufficient shear stress. Particularly at elevated temperatures, due to strong temperature dependence, the dislocation climbing rate increases dramatically, making climb no longer a minor contributor to deformation but rather a dominant mechanism that cooperates with glide to control high-temperature creep plasticity.

We fully agree with your assessment that at higher stress levels or more conventional temperatures, the shearing mechanism would undoubtedly regain dominance. Your comment is highly pertinent, and we will revise the manuscript accordingly to explicitly qualify the scope of this conclusion and emphasize its situational relevance and limitations for explaining the specific experimental phenomena observed. Once again, we deeply appreciate your valuable feedback, which has significantly enhanced the rigor and precision of our discussion.

Fig. R7 TEM micrographs after creep tests showing dislocation mixed climb mechanism at 300°C and high-temperature climb mechanism at 350°C.

Comment 9) The link between the precipitate population and the reported high-temperature strength remains largely qualitative. While the manuscript emphasizes the presence of fine L1₂ precipitates and Fe/Cr-rich clusters as the origin of the excellent high-temperature performance, there is no unified, quantitative description of their size, spacing, and volume fraction in the heat-treated and crept states, nor are these parameters incorporated into a strengthening model. The authors should quantitatively characterize the contributions of different precipitates and clusters to the high-temperature performance.

Response: Thank you for raising this important point about the quantitative link between precipitate populations and high-temperature strength. Indeed, our work currently emphasizes qualitative description, as the primary objective was to validate the feasibility of our proposed IA alloy design strategy for achieving enhanced high-temperature performance—a goal substantiated by the measured mechanical properties. This verification is based on multiple experimental results: positron annihilation spectroscopy confirming formation of vacancy-pinning solute clusters, creep experiments demonstrating the anticipated development of dislocation-pinning Cottrell atmospheres, and APT analysis revealing rare earth segregation at both particle interiors and boundaries.

Regarding your suggestion for quantitative analysis, we recognize this is a substantial

challenge. No reliable strengthening models yet exist for nanoscale clusters, and applying conventional Orowan strengthening equations would inevitably introduce significant errors. Additionally, quantifying dislocation density in aluminum alloys—materials with high stacking fault energy and inherently low dislocation density—is experimentally very difficult. Without extensive experimental data and appropriate theoretical frameworks, we believe a simplistic quantitative treatment would risk producing misleading results. Furthermore, the main message of this paper is not quantitative modeling but rather proof-of-concept for our strategy. The contributions of multi-scale microstructure to strength at room and high temperatures are illustrated in the Sankey diagram (**Fig. R8**) below, which will be included as supplementary material to help readers understand the temperature-dependent strengthening effects of each microstructural component.

Nevertheless, we agree that quantifying the effects of rare earth segregation on particle coarsening is both feasible and valuable. We are currently preparing a follow-up manuscript that will focus specifically on this aspect with the rigorous quantitative treatment it requires.

Fig. R8 Sankey diagram illustrating the contributions of multi-scale microstructural features (solute atoms, solute-vacancy clusters, precipitates, and grains) to strength at ambient and elevated

temperatures.

Comment 10) Some information is missing in both the main text and the Supporting Information. In Fig. 4d–f, no scale bars are provided, and the same issue exists in Supplementary Fig. 13, making it difficult to assess the characteristic length scales of the observed features. In addition, although the authors state that XRD was used to identify the phase constituents, the corresponding XRD patterns are not shown.

Response: Thank you for carefully reviewing our manuscript and pointing out these issues. We deeply apologize for these oversights. These errors should have been caught before submission, and we take full responsibility for this lapse in attention to detail. Regarding **Fig. 4d-f**, the dimensions were in fact labeled in the top left corner, but we recognize this may not be immediately apparent; we will therefore add clear scale bars to all panels to avoid any confusion. For **Supplementary Fig. 13**, the scale bar was indeed accidentally omitted and will be added in the revised version. In addition, we appreciate you highlighting the absence of the XRD patterns, these essential data will be added as a new supplementary figure in the Supplementary Information. We sincerely appreciate your careful review, which has helped us improve the quality and completeness of our manuscript. We assure you that all points will be thoroughly addressed in the revised version.

Responds to Reviewer #2:

This manuscript presents an “intrinsic attraction” strategy for designing heat-resistant additive manufacturing Al alloys. It employs a multi-dimensional defect confinement mechanism (vacancy trapping, dislocation pinning, precipitate stabilization), enabling the AM alloy to exhibit exceptional high-temperature performance: ~305 MPa yield strength at 300°C, ~190 MPa at 400°C, and superior creep resistance ($\dot{\epsilon} < 10^{-7}/\text{s}$) at 200–400°C. Notably, the alloy also possesses excellent processability, it's a critical concern for AM Al alloys. The manuscript is well-structured, clearly written, and the findings seem promising in the design of high-performance alloys. Thus, I would recommend it for publication after some points been addressed.

We sincerely thank the reviewer for recognizing the importance of our work and providing such detailed, constructive feedback. We have carefully addressed all the points raised, which has substantially improved the manuscript. We greatly appreciate your expertise and time in helping us enhance this work.

Comment 1) The binding energy of the two solute atoms needs to be calculated in the Al bulk instead of free states. The authors also find that there are some solute enrichments and particles form at GBs experimentally. However, there lacks evidences to demonstrate whether these solutes exhibit the similar tendency with the (co-)segregation at grain boundaries or other interfaces. Is the phenomenon of Sc occupying lattice sites within particles universal? Does Sc tend to occupy positions inside particles or at particle boundaries?

Response: Thank you for your careful reading and patient guidance. We chose to calculate binding energies in a vacuum supercell because this study aims to evaluate the intrinsic binding capability between rare earth atoms and the strengthening principal elements Fe and Cr. The crystal structure of the precipitate phase formed by Fe and Cr in aluminum differs from the FCC structure of the aluminum matrix. The binding of atoms may not only occur in the matrix, but also at the boundaries or inside of precipitated particles. Calculations in a vacuum supercell effectively eliminate interference from lattice type on atomic binding energies, thereby directly reflecting the interaction between rare earth atoms and the precipitate phase.

To clarify the interfacial segregation behavior, we have categorized the microstructure into columnar regions (CG) and fine equiaxed grain regions (FG) for detailed discussion. ①In the CG,

grain boundaries are primarily composed of an Al_6Fe eutectic network, resulting from a divorced eutectic process during solidification where Fe was rejected to the cellular boundaries of $\alpha\text{-Al}$ grains (as shown in **Fig. 3e**). However, after creep deformation, partial disintegration of the eutectic structure occurred, accompanied by coarsening of the boundary precipitates (Supplementary **Fig. 10**). To further investigate the elemental distribution in the CG after heat treatment and creep, we performed STEM-EDS analysis shown in **Fig. R9**, which clearly revealed Sc segregation inside the precipitates. The line scan results proved that the creep process promoted Sc segregation. These experimental results are presented below and will be added to the supplementary figure as Supplementary Figure 11, confirming that Sc occupying lattice positions within particles is a common phenomenon. ②In the FG, grain boundaries are predominantly decorated with $\text{Al}_{13}(\text{Fe}, \text{Cr})_{2-4}$ precipitates, which nucleate and grow along these boundaries. After creep exposure, these particles coarsen, and the grain boundaries also serve as favorable sites for the nucleation of $\text{L}_{12}\text{-Al}_3(\text{Sc}, \text{Zr})$ particles. Concurrently, Sc occupation within the $\text{Al}_{13}(\text{Fe}, \text{Cr})_{2-4}$ particles is observed, further confirming that the incorporation of Sc into particle interiors is a common behavior. It should be noted that although substitution energy calculations indicate a thermodynamic tendency for Sc atoms to occupy Al-9 sites in the $\text{Al}_{13}(\text{Fe}, \text{Cr})_4$ lattice—promoting Sc aggregation within particles—the limited availability of such sites in practice leads to the segregation of a considerable fraction of Sc at the matrix-particle interfaces (**Figs. 5d-f**). Besides the motivation from Al-9 site occupation, the intrinsic attraction between Fe/Cr and Sc atoms also comes into play. Lots of free Sc atoms in alloy matrices have been captured by Fe/Cr atoms to form nanoscale Fe/Cr-Sc clusters, because of their strong binding tendency. As well, the increased temperature or prolonged thermal exposure facilitates long-term atomic migration, allowing more Sc atoms to enter into the $\text{Al}_{13}(\text{Fe}, \text{Cr})_4$ particles. Then the thermodynamically favorable Sc substitution occurs, ultimately promoting the generation of internal Sc-rich clusters beneficial for the higher thermal stability of those particles (shown in **Figs. 5d-f** and **Supplementary Fig. 15**).

Fig. R9 Elemental distribution in the coarse-grained region (a) after heat treatment and (b) after creep; (c) and (d) show line-scan profiles of Fe, Cr, Sc, and Zr along line 1 and line 2, respectively.

Comment 2) What is the status of the alloy samples used for comparison in Figure 2a and 4a? Are all samples of comparable dimensions? Have these alloys undergone identical processing, heat treatment, or densification conditions? The authors should confirm that differences in mechanical properties stem solely from chemical composition, not from variations in porosity or residual stress. The actual alloy compositions should be specified in the text.

Response: We appreciate your careful attention to detail and your valuable suggestions for improving the manuscript. We confirm that all alloys compared in **Figs. 2a and 4a** were in an identical heat-treated state (325 °C for 4 hours) and were fabricated using the same set of optimized PBF-LB parameters, as detailed in the Methods section, to eliminate the influence of porosity and residual stress. Furthermore, the specimens for mechanical testing had consistent dimensions (as shown in the **Fig. R10**) and were sampled and processed identically. Therefore, the differences in mechanical properties can be primarily attributed to the chemical composition. The actual compositions, as verified by ICP analysis, are provided in Supplementary Table 1 and

would be specified in the revised manuscript.

Table R2 Chemical compositions of the IA alloy and comparative alloys

	Chemical composition (wt. %)					
	Fe	Cr	Ni	Sc	Zr	Al
IA alloy (Al-Fe-Cr-Sc-Zr)	2.53	2.05	/	0.65	0.32	Balance
Al-Fe-Sc-Zr	4.57	/	/	0.7	0.29	Balance
Al-Cr-Sc-Zr	/	4.52	/	0.63	0.35	Balance
Al-Ni-Sc-Zr	/	/	4.54	0.61	0.31	Balance
Al-Fe-Cr	2.49	2.04	/	/	/	Balance

Fig. R10 the dimensions of the tensile sample.

Comment 3) Figure 2 compares the mechanical properties of the IA alloy with those of other Al-based alloys. It should provide a summary table listing alloy compositions, strengthening phases, test conditions, and yield strengths to highlight the performance advantages.

Response: Thank you for this constructive suggestion. We agree that a summary table would facilitate more direct comparison and better highlight the performance advantages. In the revised manuscript, we will add a comprehensive supplementary table (**Table R3**) detailing the strengthening phases, test conditions, and yield strengths for all aluminum alloys compared in **Fig. 2**.

Table R3 Comparison of tensile properties for the IA alloy and comparative alloys (all after heat treatment, 325°C/4h) at various temperatures

Heat	strengthening	T/°C	δ0.2/MPa	UTS/MPa	elongation/%
------	---------------	------	----------	---------	--------------

		Treatment	phases				
		t					
IA	alloy	325°C 4h	Al ₁₃ (Fe,	25	492±4	530±5	12±2
(Al-2.5Fe-2Cr-Sc-Zr			Cr) ₂₋₄ ,	200	345±6	362±4	10±4
)			Al ₆ Fe,	300	275±4	305±4	7±1.5
			Al ₃ (Sc, Zr)	400	179±5	189±6	5.4±0.8
Al-4.5Fe-Sc-Zr		325°C 4h	Al ₆ Fe,	25	351±6	489±12	3.3±1.2
			Al ₁₃ Fe ₄ ,	200	257±3	340±10	5.9±0.8
			Al ₃ (Sc, Zr)	300	154±7	235±5	7.6±1
				400	147±10	165±8	4.1±0.5
Al-4.5Cr-Sc-Zr		325°C 4h	Al ₄₅ Cr ₇ ,	25	367±6	424±20	7.3±1.5
			Al ₃ (Sc, Zr)	200	247±5	294±6	8.6±0.5
				300	185±7	222±10	10±2
				400	102±4	118±12	12.9±1.5
Al-4.5Ni-Sc-Zr		325°C 4h	Al ₃ Ni,	25	269±8	440±12	6±2
			Al ₃ (Sc, Zr)	200	198±6	300±8	5±2
				300	116±2	134±12	4.2±1.5
				400	72±6	98±13	3±0.5
Al-2.5Fe-2Cr		325°C 4h	Al ₁₃ (Fe,	25	334±3	387±8	10.4±1
			Cr) ₂₋₄ ,	200	285±5	315±5	8.8±2
			Al ₆ Fe	300	235±4	255±6	7.2±2.3
				400	82±8	148±12	5.2±1.3

Comment 4) The authors attribute superior thermal stability to Fe/Cr–Sc/Zr segregation within Al₁₃(Fe, Cr)₄ precipitates. Yet, quantitative coarsening kinetics are not presented. A comparison of coarsening rates between IA and other Al-based alloys would substantiate this claim. The conclusion section mentions the particle coarsening rate at 400 °C, was not previously mention. Please provide experimental data to support this statement.

Response: Thank you for this valuable comment. You are correct that quantitative coarsening

kinetics are essential to support our claim regarding thermal stability. We have indeed conducted detailed coarsening characterization (**Fig. R11**) and calculated the particle coarsening rates (**Fig. R12**) at 400°C for the IA alloy and comparative Al alloys. To further quantify the coarsening rate of the strengthening phase, and considering its irregular morphology, this study assumes that the average particle width is approximately equivalent to the particle diameter. The classical Lifshitz–Slyozov–Wagner (LSW)¹ coarsening model was used for the analysis. The model can be expressed as:

$$dt^3 - dt_0^3 = K \cdot (t - t_0) \quad (1)$$

In this model, dt and dt_0 represent the average diameters of the secondary phase at times t and t_0 , respectively (in this study, $t_0=0$ h and $t=200$ h). K denotes the coarsening rate constant of the strengthen phase. The coarsening rates will be presented below. However, a comprehensive analysis of the coarsening mechanisms is being prepared as a separate manuscript to provide a thorough and systematic discussion. To avoid presenting incomplete data and to maintain the focus of the current paper on alloy design and strengthening mechanisms, we have removed the statement about "particle coarsening rates at 400°C" from the Conclusion section. We hope this revision is acceptable.

[1] Wu, Y. et al. Precipitate coarsening and its effects on the hot deformation behavior of the recently developed γ' -strengthened superalloys. *J. Mater. Sci. Technol.* 67, 95-104 (2021). <https://doi.org/https://doi.org/10.1016/j.jmst.2020.06.025>

Thermal exposure at 400°C for

Fig. R11 Microstructural evolution of the IA alloy and comparative Al alloys after thermal exposure at 400°C for various durations

Fig. R12 Particle width and coarsening rates at 400°C for the IA alloy and comparative Al alloys

Comment 5) The statement: “As a result, Sc/Zr atom incorporating into the L1₂ particle is reduced.

This modification remarkably slows the coarsening rates, especially compared to Al-Mg-Sc-Zr alloys” lacks supporting evidences. What are the differences between the L₁₂ particles in the IA alloy and those in Al-Mg-Sc-Zr alloys?

Response: Based on detailed microstructural characterization, we observed that in the IA alloy, the L₁₂ particles remain stable at 4~6 nm even after prolonged creep exposure, and are only about 2~4 nm following heat treatment, as shown in **Supplementary Fig. 11**. In contrast, the L₁₂ particles in the representative L₁₂-strengthened Al-Mg-Sc-Zr alloy typically exceed 5 nm after heat treatment (**Fig. R10**). We propose that the attractive interaction between Fe/Cr and Sc/Zr enhances the solubility of Sc/Zr within Al-Fe/Cr particles or clusters, thereby slowing the precipitation kinetics of L₁₂ during subsequent heat treatment or creep. Moreover, we detected the presence of Fe/Cr elements within the precipitated L₁₂ particles (**Fig.3 j**). These Fe/Cr-modified L₁₂ particles exhibit superior thermal stability compared to the conventional L₁₂ particles in the Al-Mg-Sc-Zr alloy.

[Figure Redacted]

Fig. R10 Comparison of L₁₂ precipitate size between Al-Mg-Sc-Zr and IA alloys

Comment 6) The details of calculations remain unclear, especially for solute-dislocation interaction energies. The authors claim the use of PBE in Method in the main text but LDA in Supplementary Note 2. In Page5 line5, “Here, the more negative value indicates the greater system stability.” However, the selection of elements with higher positive values in Figs 1b, d contradicts the principle of element selection. Although a negative sign appears in the

binding-energy formula, this expression may cause confusion. Please adopt a unified notation, using either negative or absolute values to represent energy.

Response: We sincerely thank you for pointing out these key issues and deeply apologize for the confusion caused by inconsistent expressions between the method section and supplementary materials, as well as the ambiguous symbol conventions of the binding energy. This is indeed an oversight during the manuscript preparation process. We hereby clarify that all DFT calculations strictly follow the methodology described in the Methods section, employing the Vienna ab-initio Simulation Package (VASP) with the generalized-gradient approximation (GGA) in the Perdew–Burke–Ernzerhof (PBE) form and projector-augmented-wave (PAW) potentials; the mention of LDA in the **Supplementary Note 2** is erroneous and will be corrected in the revised manuscript. Regarding the binding energy representation, we fully accept your suggestion and will uniformly adopt positive values to represent binding strength, thereby eliminating confusion with thermodynamic stability conventions. Following your suggestion, we have re-expressed in revised manuscript “binding energies are reported as positive quantities using $E_{\text{binding}} = E(\text{A}) + E(\text{B}) - E(\text{AB})$; a more positive value indicates stronger binding.”(as shown in page x line x). This positive-value convention is adopted because it directly and intuitively reflects the magnitude of attraction (larger values indicate stronger binding), aligns with common practice in materials science literature, and enables readers to compare segregation tendencies without sign conversion, significantly enhancing the clarity of our element selection criteria in **Figs. 1b and d**.

Comment 7) Indicate the concentration isovalue of each element in all APT reconstruction data. The manuscript alternates between “IA alloy,” “heat-resistant alloy,” and “Al-Fe-Cr-Sc-Zr alloy,” which may confuse readers. Please unify the terminology and clearly define the naming convention at the first occurrence. In Fig. 4m, is the contour value setting of the deformation charge density difference unchanged for all systems?

Response: We sincerely thank the reviewer for these insightful comments and valuable suggestions. Regarding the insufficient presentation of APT reconstruction data, we will clearly specify the concentration iso-value settings for each element in all corresponding figure captions to enhance data reliability. We deeply apologize for any confusion caused by inconsistent alloy naming throughout the manuscript and will use the term 'IA alloy' uniformly in the revised version.

When first mentioned, its composition system (Al-Fe-Cr-Sc-Zr) has been clearly defined to ensure consistency in terminology and reader friendliness. Concerning the contour value settings in Fig. 4m, we confirm that all visualization maps across different systems employ an identical iso-surface criterion ($Z_{\max}=0.04$, $Z_{\min}=-0.04$), with the unified scale bar clearly displayed on the right side of **Fig. 4m**, ensuring the objectivity of comparative analysis.